# Network properties determine neural network performance

Chunheng Jiang[1,2,5], Zhenhan Huang [1,2,5], Tejaswini Pedapati[3], Pin-Yu Chen [3], Yizhou Sun [4] & Jianxi Gao [1,2] ✉

Machine learning influences numerous aspects of modern society, empowers new technologies, from Alphago to ChatGPT, and increasingly materializes in consumer products such as smartphones and self-driving cars. Despite the vital role and broad applications of artificial neural networks, we lack systematic approaches, such as network science, to understand their underlying mechanism. The difficulty is rooted in many possible model configurations, each with different hyper-parameters and weighted architectures determined by noisy data. We bridge the gap by developing a mathematical framework that maps the neural network's performance to the network characters of the line graph governed by the edge dynamics of stochastic gradient descent differential equations. This framework enables us to derive a neural capacitance metric to universally capture a model's generalization capability on a downstream task and predict model performance using only early training results. The numerical results on 17 pre-trained ImageNet models across five benchmark datasets and one NAS benchmark indicate that our neural capacitance metric is a powerful indicator for model selection based only on early training results and is more efficient than state-of-the-art methods.

Deep neural networks (DNNs) have emerged as a crucial component of artificial intelligence (AI) and have successful applications in various domains, including computer vision, natural language processing, speech recognition, robotics, and more[1–4]. Despite these remarkable achievements, neural networks are often criticized as black boxes and remain challenging to comprehend due to their nonlinear and complex nature[5]. Increasing research is developing more interpretable DNN architectures, such as those based on attention mechanisms or interpretable features[6–8]. Nevertheless, neural network training is complex and affected by various factors such as noisy training data, neural architecture, loss function, and optimization algorithms, remaining a critical challenge to uncover the black box of DNNs[9,10].

The training process is an iterative update of the synaptic connection weights[11,12]. The straightforward way is to model the process as a discrete dynamical system, which provides a theoretical foundation for analyzing convergence rates and generalization error bounds[13–16].

However, existing approaches have primarily focused on the macroscopic and collective behavior of neurons in neural networks[17–19], without explicitly examining the individual interactions between trainable weights or synaptic connections and their co-evolution during training.

Transfer learning is a widely used and effective technique in deep learning that leverages pre-trained models to solve numerous complex problems. One application is the large language model ChatGPT, which is well-versed in using transfer learning for question answering[20,21]. However, selecting the optimal pre-trained model for a given task remains challenging because thoroughly training each candidate is computationally expensive and time-consuming, promoting an urgent need for an efficient predictive measure based only on early training results.

A comprehensive understanding of neural dynamics is the critical step to addressing these challenges, ultimately leading to optimal

[1]Network Science and Technology Center, Rensselaer Polytechnic Institute, Troy, NY, USA. [2]Department of Computer Science, Rensselaer Polytechnic Institute, Troy, NY, USA. [3]IBM Thomas J. Watson Research Center, Yorktown Heights, NY, USA. [4]Department of Computer Science, University of California, Los Angeles, CA, USA. [5]These authors contributed equally: Chunheng Jiang, Zhenhan Huang. ✉e-mail: gaoj8@rpi.edu

neural network design. We fill the gap by adopting a microscopic perspective to investigate the edge dynamics of synaptic connections induced by stochastic gradient descent (SGD)[11] through differential equations. The proposed new approach forms an associated network of edges and models neural network training as a networked dynamical system over these edges. However, solving the nonlinear networked edge dynamics poses significant computational challenges, given the millions of weights in convolutional neural networks, such as MobileNet[22] (16 millions of weights) and VGG16[1] (528 millions of weights). To overcome this limitation, we use the network reduction approach (GBB reduction) proposed by Gao et al. to decouple the neural network system, which enables us to map the neural network's performance to its network characters[23,24]. Our analysis advances several critical problems in AI, such as learning curve prediction, model selection, and zero-shot learning. Specifically, our universal approach significantly improves the relative ranking prediction of pre-trained models by 9.1% to 65.3% using early training statistics from as few as five epochs. These findings demonstrate the effectiveness of our framework in finding the best predictive model and have significant implications for neural network architecture design and search in various applications.

## Results

### Map from a neural network to an associated graph of edges

The critical step is to map an artificial neural network to a networked dynamical system so that we can use the corresponding approaches to analyze them. We built a mapping scheme $\phi: G_A \mapsto G_B$, from a neural network $G_A$ to an associated graph $G_B$. The topology of the edges (synaptic connections) follows a well-defined line graph proposed by Nepusz and Vicsek[25], and nodes of $G_B$ are edges of $G_A$. More precisely, each node in $G_B$ is associated with a trainable parameter in $G_A$. For an MLP, each edge has a trainable weight, and the edge set of $G_A$ is also the synaptic connection of $G_B$. For a CNN, this one-to-one mapping from neurons on layer $\ell$ to layer $\ell + 1$ is replaced by a one-to-many mapping because of weight sharing, e.g., a parameter in a convolutional filter is repeatedly used in forward propagation and associated with multiple pairs of neurons from the two neighboring layers. Since the error gradients flow in a reversed direction, we reverse the corresponding links of the proposed line graph for $G_B$. Specifically, given any pair of nodes in $G_B$, if they share an associated intersection neuron in FP propagation routes, a link with a reversed direction will be created for them. Fig. 1a demonstrates the mapping of an example MLP. We have the topology of $G_B$ in place, but the weights of links in $G_B$ are not yet specified. To make up for these missing components, we reveal the interactions of synaptic connections from SGD, quantify the interaction strengths and then define the weights of links in $G_B$ accordingly (see Methods section for detailed derivation).

Figure 1b shows how to use our approach to predict the performance of a pre-trained neural network model based on transfer learning. The output layer of each pre-trained model is replaced with a three-layer neural capacitance probe (NCP) unit with (1) a dense layer of size 256 and (2) a dense layer of size 128. Each of these dense layers follows (3) a batch normalization[26], and (4) is followed by a dropout layer with a dropout probability of 0.4. Before fine-tuning, we initialize the NCP unit using Kaiming Normal initialization[27]. See Supplementary Note 3 for details about the three layers in NCP.

### Neural network model selection with the neural capacitance $\beta_{eff}(t)$

We evaluate 17 pre-trained ImageNet models implemented in Keras[28], including AlexNet, VGGs (VGG16 & 19), ResNets (ResNet50, 50V2, 101, 101V2, 152, 152V2), DenseNets (DenseNet121, 169, 201), MobileNets (MobileNet & MobileNetV2), Inceptions (InceptionV3 & InceptionResNetV2) and Xception, to measure the performance of our approach. Furthermore, we used four benchmark datasets, CIFAR10, CIFAR100,

SVHN, Fashion MNIST of size $32 \times 32 \times 3$, and one Kaggle challenge dataset, Birds of size $224 \times 224 \times 3$, and split the original train/test. Also, 15K original training samples are set aside to validate our approach on each dataset. We set a batch size of 64 and a learning rate of 0.001, fine-tuning each modified pre-trained model for $T = 50$ epochs. As shown in Algorithm 1, the NCP does not involve fine-tuning and is merely used to calculate the neural capacitance $\beta_{eff}(t)$, which varies as the number of epochs $t$ changes. To keep the notation succinct, we use $\beta_{eff}$ to represent $\beta_{eff}(t)$. According to Theorem 1 (see Methods section on the property of the neural capacitance), when the model converges, $\beta_{eff} \to 0$. Indirectly, the model's predictability can be determined by the relation between the training $\beta_{eff}$ and the validation accuracy $I$. Since both $\beta_{eff}$ and $I$ are available during fine-tuning, we collect a set of data points of these two in the early phase as the observations and fit a regularized linear model $I = h(\beta_{eff}; \boldsymbol{\theta})$ with Bayesian ridge regression[29], where $\boldsymbol{\theta}$ are the associated coefficients. The estimated predictor $I = h(\beta_{eff}; \theta^*)$ makes prediction of the final accuracy of models by setting $\beta_{eff} = 0$, i.e., $I^* = h(0; \theta^*)$, see Fig. 1c an example in row 3 of Fig. 2. One can either retain or remove the NCP and fine-tune the selected model to fully train the best model.

To control the randomness, we repeat 20 times of the fine-tuning experiments for each model and analyze the average result. As shown in Fig. 2, the pre-trained models are converged after the fine-tuning on CIFAR10. For each model, we collect the validation accuracy (blue stars in row 1) and $\beta_{eff}$ on the training set (green squares in row 2) during the early stage of fine-tuning as the observations (e.g., green squares in row 3 marked by the green box for five epochs), then use these observations to predict the test accuracy unseen before the fine-tuning terminates. The blue lines are estimated $h(\cdot; \boldsymbol{\theta})$, the true test accuracy at $T$ and the predicted accuracy are marked as red triangles and blue stars, respectively. Both the estimates and predictions are accurate. For better illustration, learning curves are visualized on a log scale.

The relative rank of these candidates is more important than their exact values of predicted accuracy in model selection. Thus, we choose Spearman's rank correlation coefficient $\rho$ to evaluate and compare different approaches. We calculate $\rho$ over the ground truth test accuracy at epoch $T$ and all pre-trained models' predicted accuracy $I^*$. In Fig. 3a, we report the ground truth and predicted accuracy for each model on CIFAR10, as well as the overall ranking performance measured by $\rho$. It indicates that $\beta$-based ranking is reliable with $\rho > 0.9$. We also report the complete results on all five datasets in Fig. 4. The numerical results indicate that the approach is general for different datasets.

The estimation quality of $h$ determines how well the relation between $I$ and $\beta_{eff}$ is captured. Besides the regression method, the starting epoch $t_0$ of the observations also plays a role in the estimation. As shown in Fig. 3b, we evaluate the impact of $t_0$ on $\rho$ of our approach. As expected, when fixing the length of learning curves, a higher $t_0$ usually produces a better $\rho$. Since our ultimate goal is to predict with the early observations, $t_0$ should also be constrained to a small value. To make the comparisons fair, we view $t_0$ as a hyper-parameter, and select it according to the Bayesian information criterion (BIC)[30], as shown in row 3 of Fig. 2.

### Impact of size of training set

It is important to understand scalability and the performance sensitivity to training set sizes. Thus, we further split the CIFAR10, which has 50K original training and 10K testing samples, into 35K for training and 15K for validation. In studying the dynamics of neural network training, it is essential to understand how varying the training size influences the effectiveness of our approach. We select the first {10,15,20,25,30}K of the original 50K samples as the reduced-size training set and the last 10K samples as the validation set to fine-tune the pre-trained models for 50 epochs. As shown in Fig. 3c, we can use a training set of size as small as 25K to achieve similar performance to that uses all 35K training

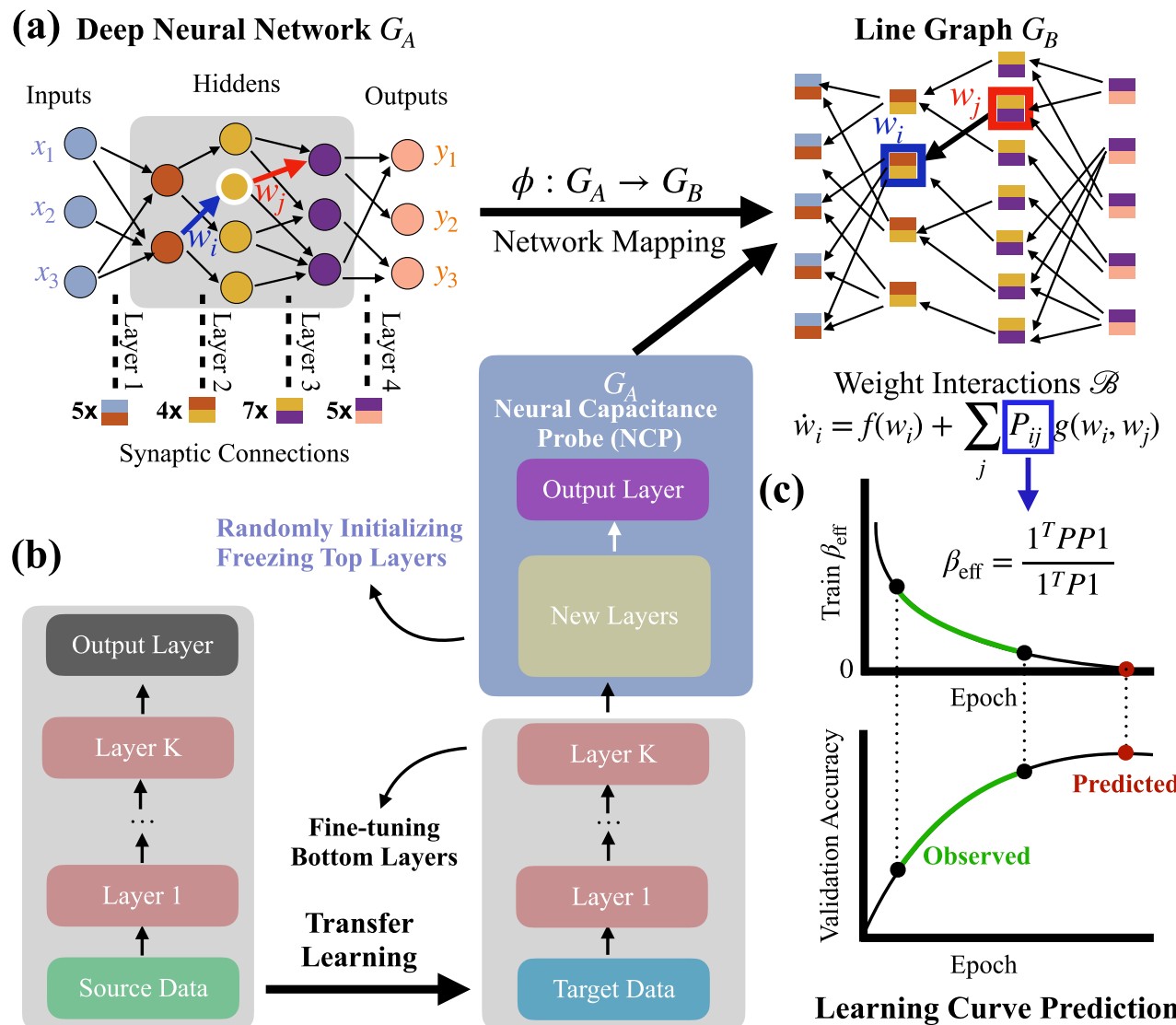

**Fig. 1 | Illustration of our framework. a** An example multilayer perceptron (MLP) $G_A$ is mapped to a directed line graph $G_B$, which is governed by an edge dynamics $\mathcal{B}$. Each node (dichromatic square) of $G_B$ is associated with a synaptic connection linking two neurons (in different colors) from different layers of $G_A$. **b** A diagram of transfer learning from the source domain (left stack) to a target domain (right stack). The pre-trained model is modified by adding additional layers, i.e., installing a neural capacitance probe (NCP) unit, on top of the bottom layers. The NCP is frozen with a set of randomly initialized weights, and only the bottom layers are

fine-tuned. **c** Observed partial learning curves (green line segments) of validation accuracy over the early-stage training epochs and the corresponding neural capacitance metric $\beta_{\text{eff}}$ during fine-tuning. The predicted final accuracy at $\beta_{\text{eff}} \to 0$ (red dot) is used to select the best one from a set of models. The metric $\beta_{\text{eff}}$ relies on $G_B$'s weighted adjacency matrix $P$, which itself is derived from the reformulation of the training dynamics. To predict the performance, a lightweight $\beta_{\text{eff}}$ of the NCP is used instead of the heavyweight one over the entire network on the right stack of (**b**).

samples. It has an important implication for efficient neural network training, because the size of the required training set can be significantly reduced (around 30% in our experiment) while maintaining similar model ranking performance. Note that the true test accuracy used in computing $\rho$ is the same test accuracy for the model trained from 35K training samples and it's shared by all the five cases {10,15,20,25,30}K in our analysis.

### Comparison with other approaches

For comparison analysis, we considered two families of predictors: learning curve (LC) based predictors, and transferability measures (TMs) as the baselines. (i) LC predictors. Chandrashekaran and Lane[31] treated the current LC as an affine transformation of previous LCs. They built an ensemble of transformations employing previous LCs and the first few epochs of the current LC to predict the final accuracy

of the current LC. Baker et al.[32] proposed an SVM-based LC predictor using features extracted from previous LCs, including the architecture information such as the number of layers, parameters, and training techniques such as learning rate and learning rate decay. A separate SVM is used to predict the accuracy of an LC at a particular epoch. Domhan et al.[33] trained an ensemble of parametric functions that observe the first few epochs of an LC and extrapolate it. Klein et al.[34] devised a Bayesian neural network to model the functions that Domhan formulated to capture the structure of the LCs more effectively. Wistuba and Pedapat[35] trained a transfer learning-based predictor on LCs generated from other datasets. It is a neural network-based predictor that leverages architecture and dataset embedding to capture the similarities between the architectures of various models and also the other datasets that it was trained on. (ii) Transferability measures. As an alternative estimation of the final performance of neural network

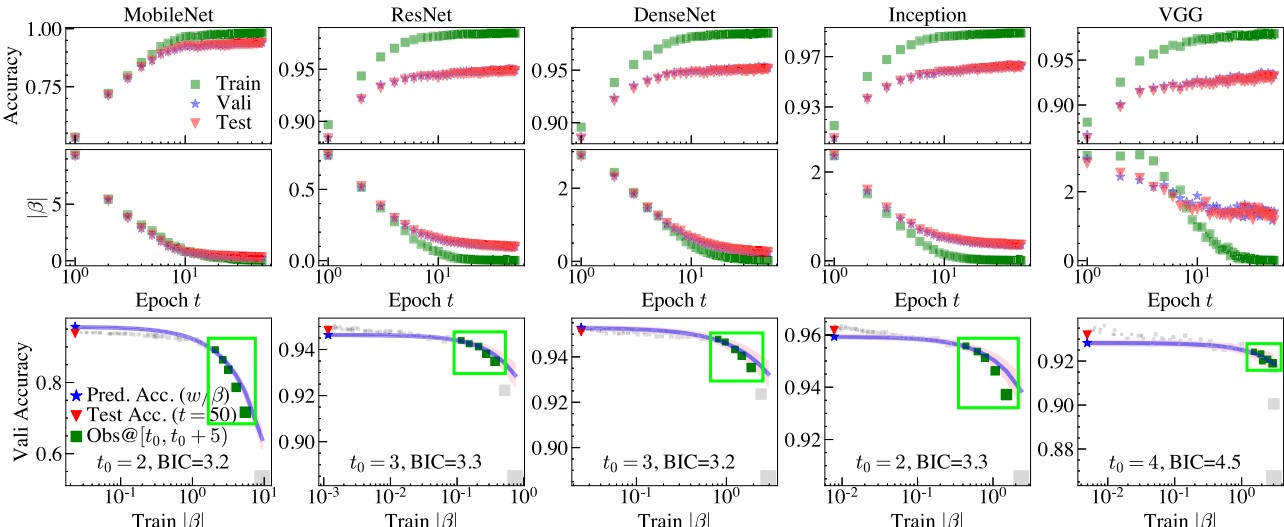

**Fig. 2 | Learning curves of five representative pre-trained models. $\beta_{\text{eff}}$.** The first row shows the Accuracy as a function of Epoch $t$ and the second row denotes the $\beta$ as a function of Epoch $t$. A regularized linear model $h(\cdot;\boldsymbol{\theta})$ (blue curve in row 3) is estimated with Bayesian ridge regression using a few of observations of $\beta_{\text{eff}}$ on training set and validation accuracy $l$ during early fine-tuning. The starting epoch $t_0$ of observations affects the fit of $h$, and is automatically determined according to BIC, and the true test accuracy at epoch 50 is predicted with $l = h(0; \boldsymbol{\theta}^*)$.

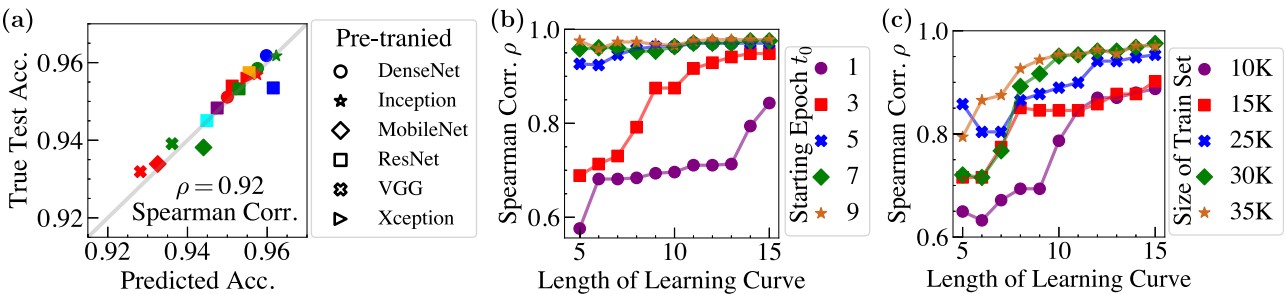

**Fig. 3 | The sensitivity analysis of the neural capacitance's predictive capability. a** Our $\beta_{\text{eff}}$ based prediction of the validation accuracy versus the true test accuracy at epoch 50 of seven representative pre-trained models. Each shape is associated with one type of pre-trained models. Distinct models of the same type are marked in different colors. Because the accuracy of AlexNet is much lower than others, we exclude it for better visualization. Its predicted accuracy is 0.871, and the true test accuracy is 0.868. If it is included, $\rho = 0.93 > 0.92$. **b** Impacts of the starting epoch $t_0$ of the observations and (**c**) the number of training samples on the ranking performance of our $\beta_{\text{eff}}$ based approach.

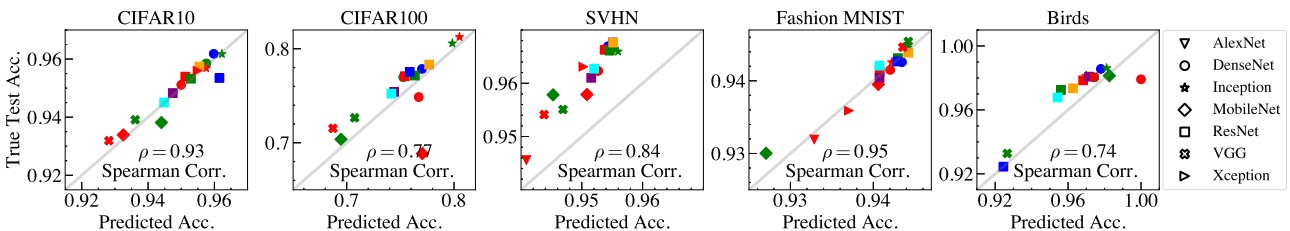

**Fig. 4 | The validation accuracy prediction of pre-trained models on all five datasets.** The validation accuracy based on $\beta_{\text{eff}}$ is strongly correlated with the true test accuracy of these models after fine-tuning for $T = 50$ epochs. The Spearman's ranking correlation $\rho$ is used to quantify the performance in model selection. Each shape is associated with one type of pre-trained models. Distinct models of the same type are marked in different colors. To be noted, each includes AlexNet in computing $\rho$s.

models, some transferability measures (TMs) are developed[36–47], and many of them are training-free metrics for assessing the performance of neural networks. Notably, our approach has access to some observations collected from early training, and therefore our prediction mechanism is more similar to the learning curve prediction than those TM-based approaches that are designed as a surrogate of the transferability without fine-tuning or re-training. In addition to LC-based predictors, we compared our method with training-free NAS methods. The result is shown in the Supplementary Note 8. Direct comparison

on the prediction performance (indicated by the ranking correlation) is not desirable since training-free NAS methods do not require training while our proposed method requires training of the model to compute $\beta_{\text{eff}}$.

We select several LC predictors, such as two heuristic rules the last seen value (LSV)[48] and the best-seen value (BSV), BGRN[32], CL[31], as well as three representative TMs: NCE[36], LEEP[37] and LogME[38] as the baselines. As shown in Table 1 and Supplementary Fig. S1, using a few observations, e.g., only 5 epochs, our approach can achieve from 9.1%

**Table 1 | A comparison between ours and the baselines in model ranking**

| Dataset | CIFAR10 | | CIFAR100 | | SVHN | | Fashion MNIST | | Birds | |
|---|---|---|---|---|---|---|---|---|---|---|
| *LLC* | 5 | 10 | 5 | 10 | 5 | 10 | 5 | 10 | 5 | 10 |
| Ours | **0.93** | **0.98** | 0.77 | 0.80 | **0.84** | **0.88** | **0.95** | **0.89** | **0.74** | **0.79** |
| BSV | 0.86 | 0.89 | 0.55 | 0.80 | 0.74 | 0.78 | 0.53 | 0.60 | 0.52 | 0.61 |
| LSV | 0.85 | 0.87 | 0.55 | 0.80 | 0.73 | 0.70 | 0.49 | 0.45 | 0.48 | 0.45 |
| BGRN | 0.74 | 0.78 | 0.45 | 0.60 | 0.63 | 0.65 | 0.57 | 0.59 | 0.53 | 0.52 |
| LC | 0.85 | 0.85 | 0.50 | 0.58 | 0.44 | 0.10 | 0.55 | 0.61 | 0.50 | – |
| LogME | 0.593 | | 0.716 | | −0.400 | | 0.010 | | 0.132 | |
| LEEP | 0.635 | | 0.593 | | 0.338 | | 0.159 | | −0.243 | |
| NCE | 0.743 | | **0.816** | | 0.152 | | −0.029 | | 0.049 | |
| *Imprv (%)* | 9.1 | 10.2 | −5.7 | −2.0 | 12.4 | 13.3 | 65.3 | 49.2 | 40.1 | 30.6 |

The notation *LLC* represents the length of the learning curve, and *Imprv* represents the relative improvement of our approach to the best baseline. The TMs are evaluated based on https://github.com/thuml/LogME repository. Due to the failure of the https://github.com/tdomhan/pylearningcurvepredictor supporting package of LC, there is a missing $\rho$ at *LLC* of 10, which does not affect our conclusions.

up to 65.3% relative improvements over the best baseline on CIFAR10, SVHN, Fashion MNIST, and Birds. On CIFAR100, NCE achieves marginally better performance than ours with 10 observations. Moreover, since each pre-trained model produces one learning curve per run, we also report our ranking performance and the baselines based on learning curves collected in individual runs (Supplementary Fig. S2).

**Running time analysis**

Our approach is efficient, especially for large and deep neural networks. Different from the training task that involves a full FP and BP, i.e. $T_{train} = T_{FP} + T_{BP}$, computing $\beta_{eff}$ only requires to compute the adjacency matrix $P$ according to Eq.(7) on the NCP unit, $T_{\beta_{eff}} = T_{NCP}$. Although the computation is complicated, the NCP is lightweight. The computing cost per epoch is comparable to the training time per epoch (see Supplementary Fig. S3). Let $T_{\beta_{eff}} = c \times T_{train}$. If $c > 1$, i.e., $T_{\beta_{eff}}$ is higher than $T_{train}$, vice versa. Considering the required epochs, our approach needs $k$ observations, and takes $T_{ours} = k \times T_{\beta_{eff}}$. To obtain the ground-truth final accuracy by running $K$ epochs, it takes $T_{full} = K \times T_{train}$. If $T_{full} > T_{ours}$, our $\beta_{eff}$ based prediction is cheaper than "just training longer". It indicates that $K \times T_{train} - k \times T_{\beta_{eff}} = (K - c \times k) \times T_{train} > 0$, saving us $K - c \times k$ more training epochs.

We perform a running time analysis of the two tasks with $4 \times$ NVIDIA Tesla V100 SXM2 32GB, and visualize the related times in Supplementary Fig. S3. On average $c = T_{\beta_{eff}} / T_{train} \approx 1.3$, computing $\beta_{eff}$ takes 1.3 times of the training per epoch. But the efforts are paying off, as we can predict the final accuracy by observing only $k = 10$ of $K = 100$ full training epochs, $T_{ours}$ is only 13% of $T_{full}$. When the observations are used for LC prediction, the heuristics directly take one observation (last or best) as the predicted value, so they are mostly computationally cheap but have sub-optimal model ranking performances. BGRN and CL require more computational time because both need training a predictor with a set of full learning curves from other models. Our approach also estimates a predictor but does not need any external LCs. Next, we assume that each model only observes $k = 5$ epochs and conduct a running time analysis of these approaches over LC prediction, including estimating a predictor. As shown in Supplementary Table S1, our approach applies Bayesian ridge regression to efficiently estimate the predictor $l = h(\beta_{eff}; \boldsymbol{\theta})$, taking comparable time as BGRN, significantly less than CL. Nevertheless, it performs best in model ranking. In contrast, the most expensive CL, does not perform well, sometimes even worse.

## Discussion

In Network Science, a fundamental objective is to comprehend the functioning of a network based on its structure with broad applications in many fields. This work attempts to advance our understanding of the functioning of artificial neural networks through a grasp of complex networks. Recently, some prior works explore the neural network SGD training dynamics, regarding the global convergence[49], system identification[50,51], as well as deep neural network generalization[52]. For example, Goldt et al.[53] formulated the SGD dynamics of over-parameterized two-layer neural networks with a set of differential equations. Furthermore, some exciting phenomena[54] emerge during the early phase of neural network training, such as trainable sparse sub-networks emerge[55], gradient descent moves into a small subspace[56]. Moreover, there exists a critical effective connection between layers[57]. Inspired by the insights gained from studying the neural network training dynamics through a networked dynamical systems lens, we developed a theoretically sound framework for improving neural network model selection.

Our work presents a novel perspective of neural network model selection by directly exploring the dynamical evolution of synaptic connections during neural network training. Our framework reformulates SGD-based neural network training dynamics as an edge dynamics $\mathcal{B}$ to capture the mutual interaction and dependency of synaptic connections. Accordingly, a networked system is built by converting a neural network $G_A$ to a line graph $G_B$ with the governing dynamics $\mathcal{B}$, which induces a definition of the link weights in $G_B$. Moreover, a topological property $\beta_{eff}$ of $G_B$ is developed and shown to be an effective metric in predicting the ranking of a set of pre-trained models based on early training results.

There are several important directions that we intend to explore in the future, including: (i) Simplify the adjacency matrix $P$ to capture the dependency and mutual interaction between synaptic connections, e.g., approximate gradients using local information[58], (ii) extend the proposed framework to more neural architecture search (NAS) benchmarks[59–62] to select the best subnetwork, and (iii) design an efficient algorithm to optimize neural network architectures directly.

## Methods

**Dimension reduction of networked systems**

Real-world complex systems, such as plant-pollinator interactions[63] and the spread of COVID-19[64], are commonly modeled using networks[65,66]. Consider a network $G = (V, E)$ with nodes $V$ and edges $E$. Let $n = |V|$ be the number of nodes in the network, the interactions between nodes can be formulated as a set of differential equations

$$\dot{x}_i = f(x_i) + \sum_{j \in V} P_{ij} g(x_i, x_j), \forall i \in V, \tag{1}$$

where $x_i$ is the state of node $i$ in the system. For instance, in an ecological network, $x_i$ could represent the abundance of a particular species of plant, while in an epidemic network, it could represent the infection rate of a person. The adjacency matrix $P$ encodes the

interaction strength between nodes, where $P_{ij}$ is the entry in row $i$ and column $j$. The functions $f(\cdot)$ and $g(\cdot,\cdot)$ capture the internal and external impacts on node $i$, respectively. Typically, these functions are nonlinear. Let $x = (x_1, x_2, ..., x_n)$. For a small network, given an initial state, one can run a forward simulation for an equilibrium state $x^*$, such that $\dot{x}_i^* = f(x_i^*) + \sum_{j \in V} P_{ij} g(x_i^*, x_j^*) = 0$.

However, when the size of the system goes up to millions or even billions, it will pose a big challenge to solve the coupled differential equations. The problem can be efficiently addressed by employing a mean-field technique[23,24], where a linear operator $\mathcal{L}_P(\cdot)$ is introduced to decouple the system. Specifically, $\mathcal{L}_P$ depends on the adjacency matrix $P$ and is defined as $\mathcal{L}_P(z) = \frac{1^T P z}{1^T P 1}$, where $z \in \mathcal{R}^n$. Let $\delta_{\text{in}} = P1$ and $\delta_{\text{out}} = 1^T P$ be the in- and out-degrees of nodes. For a weighted $G$, the degrees are weighted as well. Applying $\mathcal{L}_P(\cdot)$ to $\delta_{\text{in}}$, it gives

$$\beta_{\text{eff}} = \mathcal{L}_P(\delta_{\text{in}}) = \frac{1^T P \delta_{\text{in}}}{1^T \delta_{\text{in}}} = \frac{\delta_{\text{out}}^T}{\delta_{\text{in}}} 1^T \delta_{\text{in}}, \qquad (2)$$

which proves to be a powerful metric to measure the resilience of networks, and has been applied to make reliable inferences from incomplete networks[67,68]. We use it to measure the predictive ability of a neural network, whose training in essence is a dynamical system. For an overview of the related technique, see Supplementary Note 6.

## Neural network training is a dynamical system

Conventionally, training a neural network is a nonlinear optimization problem. Because of the hierarchical structure of neural networks, the training procedure is implemented by two alternate procedures: forward-propagation (FP) and back-propagation (BP), as described in Fig. 1a. During FP, data goes through the input layer, hidden layers, up to the output layer, which produces the predictions of the input data. The differences between the outputs and the labels of the input data are used to define an objective function $\mathcal{C}$, a.k.a training error function. BP proceeds to minimize $\mathcal{C}$, in a reverse way as did in FP, by propagating the error from the output layer down to the input layer. The trainable weights of synaptic connections are updated accordingly.

Let $G_A$ be a neural network, $w$ be the flattened weight vector of $G_A$, and $z$ be the activation values. As a whole, the training of a neural network $G_A$ can be described with two coupled dynamics: $\mathcal{A}$ on $G_A$, and $\mathcal{B}$ on $G_B$, where nodes in $G_A$ are neurons, and nodes in $G_B$ are the synaptic connections. The coupling relation arises from the strong inter-dependency between $z$ and $w$: the states $z$ (activation values or activation gradients) of $G_A$ are the parameters of $\mathcal{B}$, and the states $w$ of $G_B$ are the trainable parameters of $G_A$. If we put the whole training process in the context of networked systems, $\mathcal{A}$ denotes a *node dynamics* because the states of nodes evolve during FP, and $\mathcal{B}$ expresses an *edge dynamics* because of the updates of edge weights during BP[13,69,70]. Mathematically, we formulate the node and edge dynamics based on the gradients of $\mathcal{C}$:

$$(\mathcal{A}) \; dz/dt \approx h_{\mathcal{A}}(z, t; w) = -\nabla_z \mathcal{C}(z(t)), \qquad (3)$$

$$(\mathcal{B}) \; dw/dt \approx h_{\mathcal{B}}(w, t; z) = -\nabla_w \mathcal{C}(w(t)), \qquad (4)$$

where $t$ denotes the training step. Let $a_i^{(\ell)}$ be the pre-activation of node $i$ on layer $\ell$, and $\sigma_\ell(\cdot)$ be the activation function of layer $\ell$. Usually, the output activation function is a softmax. The hierarchical structure of $G_A$ exerts some constraints over $z$ for neighboring layers, i.e., $z_i^{(\ell)} = \sigma_\ell(a_i^{(\ell)}), 1 \le i \le n_\ell, \forall 1 \le \ell < L$ and $z_k^{(L)} = \exp\{a_k^{(L)}\}/\sum_j \exp\{a_j^{(L)}\}, 1 \le k \le n_L$, where $n_\ell$ is the total number of neurons on layer $\ell$, and $G_A$ has $L+1$ layers. It also presents a dependency between $z$ and $w$, e.g, when $G_A$ is an MLP without bias,

$a_i^{(\ell)} = w_i^{(\ell)T} z^{(\ell-1)}$, which builds a connection from $G_A$ to $G_B$. It is obvious, given $w$, the activation $z$ satisfying all these constraints, is also a fixed point of $\mathcal{A}$. Meanwhile, an equilibrium state of $\mathcal{B}$ provides a set of optimal weights for $G_A$.

The metric $\beta_{\text{eff}}$ is a universal metric to characterize different types of networks, including biological neural networks[71]. Because of the generality of $\beta_{\text{eff}}$, we analyze how it looks on artificial neural networks, which are designed to mimic the biological counterparts for general intelligence. Therefore, we set up an analog system for the trainable weights. To the end, we build a line graph for the trainable weights, and reformulate the training dynamics in the same form as the general dynamics (Eq. (1)). The reformulated dynamics reveals a simple yet powerful property regarding $\beta_{\text{eff}}$, which is utilized to predict the final accuracy of $G_A$ with a few observations during the early phase of the training.

## Quantify the interaction strengths of edges

In SGD, each time a batch of samples is chosen to update $w$, i.e., $w \leftarrow w - \alpha \nabla_w \mathcal{C}$, where $\alpha > 0$ is the learning rate. When desired conditions are met, training is terminated. Let $\delta^{(\ell)} = [\partial \mathcal{C}/\partial z_1^{(\ell)}, \cdots, \partial \mathcal{C}/\partial z_{n_\ell}^{(\ell)}]^T \in \mathcal{R}^{n_\ell}$ (in some literature $\delta^{(\ell)}$ is defined as gradients with respect to $a^{(\ell)}$, which does not affect our analysis) be the activation gradients, and $\sigma_\ell' = [\sigma_{\ell,1}', \cdots, \sigma_{\ell,n_\ell}']^T \in \mathcal{R}^{n_\ell}$ be the derivatives of activation function $\sigma$ for layer $\ell$, with $\sigma_{\ell,k}' = \sigma_\ell'(a_k^{(\ell)}), 1 \le k \le n_\ell, 1 \le \ell \le L$. To understand how the weights $W^{(\ell)}$ affect each other, we explicitly expand $\delta^{(\ell)}$ and have $\delta^{(\ell)} = W^{(\ell+1)T}(W^{(\ell+2)T}(\cdots(W^{(L-1)T}(W^{(L)T}(z^{(L)} - y)) \odot \sigma_{L-1}') \cdots) \odot \sigma_{\ell+2}') \odot \sigma_{\ell+1}')$, where $\odot$ is the Hadamard product. We find that $W^{(\ell)}$ is associated with all accessible parameters on downstream layers, and the recursive relation defines a high-order hyper-network interaction[72] between any $W^{(\ell)}$ and the other parameters. With the fact that $x \odot y = \Lambda(y)x$, where $\Lambda(y)$ is a diagonal matrix with the entries of $y$ on the diagonal, we have $\delta^{(\ell)} = W^{(\ell+1)T} \Lambda(\sigma_{\ell+1}') \delta^{(\ell+1)} = W^{(\ell+1)T} \Lambda(\sigma_{\ell+1}') W^{(\ell+2)T} \Lambda(\sigma_{\ell+2}') \cdots W^{(L-1)T} \Lambda(\sigma_{L-1}') W^{(L)T}(z^{(L)} - y)$. For a ReLU $\sigma_\ell(\cdot)$, $\sigma_\ell'$ is binary depending on the sign of the input pre-activation values $a^{(\ell)}$ of layer $\ell$. If $a_i^{(\ell)} \le 0$, then $\sigma_\ell'(a_i^{(\ell)}) = 0$, blocking a BP propagation route of the prediction deviations $z^{(L)} - y$ and giving rise to *vanishing gradients*.

We intended to build direct interactions between synaptic connections. It can be done by identifying which units provide direct physical interactions to a given unit and appear on the right-hand side of its differential equation $\mathcal{B}$ in Eq.(3), and how much such interactions come into play. There are multiple routes to build up a direct interaction between any pair of network weights from different layers, as presented by the product terms in $\delta^{(\ell)}$. However, the coupled interaction makes it an impossible task, which is well-known as a *credit assignment problem*[51,73]. We propose a remedy. The impacts of all the other units on $W^{(\ell)}$ is approximated by direct, local impacts from $W^{(\ell+1)}$, and the others' contribution as a whole is encoded in the activation gradient $\delta^{(\ell+1)}$. Moreover, we have the weight gradient (Supplementary Note 1)

$$\nabla_{W^{(\ell)}} = \Lambda(\sigma_\ell') \delta^{(\ell)} z^{(\ell-1)T} = \Lambda(\sigma_\ell') W^{(\ell+1)T} \Lambda(\sigma_{\ell+1}') \delta^{(\ell+1)} z^{(\ell-1)T}, \qquad (5)$$

which shows the dependency of $W^{(\ell)}$ on $W^{(\ell+1)}$, and itself can be viewed as an explicit description of the dynamical system $\mathcal{B}$ in Eq.(3). Put it in terms of a differential equation, we have

$$\frac{dW^{(\ell)}}{dt} = -\Lambda(\sigma_\ell') W^{(\ell+1)T} \Lambda(\sigma_{\ell+1}') \delta^{(\ell+1)} z^{(\ell-1)T} \triangleq F(W^{(\ell+1)}). \qquad (6)$$

Because of the mutual dependency of the weights and the activation values, it is hard to make an exact decomposition of the impacts of different parameters on $W^{(\ell)}$. But, in the gradient $\nabla_{W^{(\ell)}}$, $W^{(\ell+1)}$ presents as

an explicit term and contributes the direct impact on $W^{(\ell)}$. To capture such direct impact and derive the adjacency matrix $P$ for $G_B$, we apply Taylor expansion on $\nabla_{W^{(\ell)}}$ and have

$$P^{(l,l+1)} = \partial^2 \mathcal{C} / \partial W^{(\ell)} \partial W^{(\ell+1)}, \tag{7}$$

which defines the interaction strength between each pair of weights from layer $\ell + 1$ to layer $\ell$. For a detailed derivation of $P$ on MLP and general neural networks, see Supplementary Notes 2 and 3. Let $w = (w_1, w_2, \ldots)$ be a flattened vector of all trainable weights of $G_A$. Given a pair of weights $w_i$ and $w_j$, one from layer $\ell_1$, another from layer $\ell_2$. If $\ell_2 = \ell_1 + 1$, the entry $P_{ij}$ is defined according to Eq.(7), otherwise $P_{ij} = 0$. Considering the scale of the trainable parameters in $G_A$, $P$ is very sparse. Let $W^{(\ell+1)*}$ be the equilibrium states (Supplementary Note 3), the training dynamics Eq.(6) is reformulated into the form of Eq.(1), and gives the edge dynamics $\mathcal{B}$ for $G_B$:

$$\dot{w}_i = f(w_i) + \sum_j P_{ij} g(w_i, w_j), \tag{8}$$

with $f(w_i) = F(w_i^*)$ and $g(w_i, w_j) = w_j - w_j^*$. The value of weights at an equilibrium state $\{w_j^*\}$ is unknown, but it is a constant and does not affect the computing of $\beta_{\text{eff}}$.

## Property of the neural capacitance

According to Eq.(7), we have the weighted adjacency matrix $P$ of $G_B$ in place. The matrix $P$ encodes rich information of the network, such as the topology, the weights, the gradients, and the training labels indirectly. Now we quantify the total impact that a trainable parameter (or synaptic connection) receives from itself and the others, corresponding to the weighted in-degrees $\delta_{\text{in}} = P\mathbf{1}$. Applying $\mathcal{L}_P(\cdot)$ to $\delta_{\text{in}}$, we get a "counterpart" metric $\beta_{\text{eff}} = \mathcal{L}_P(\delta_{\text{in}})$ to measure the predictive ability of a neural network $G_A$, as the resilience metric (Eq. (2)) does to a general network $G$. If $G_A$ is an MLP, we can explicitly write the entries of $P$ and $\beta_{\text{eff}}$. For details of how to derive $P$ and $\beta_{\text{eff}}$ of an MLP, see Supplementary Note 2. Moreover, we prove in Theorem 1 below that as $G_A$ converges, $\nabla_W^{(\ell)}$ vanishes, and $\beta_{\text{eff}}$ approaches zero (see Supplementary Note 4).

**Theorem 1.** Let ReLU be the activation function of $G_A$. When $G_A$ converges, then $\beta_{\text{eff}} = 0$.

To be noted that a small value is added to the denominator of Eq.(2) to avoid a possible 0/0.

**Algorithm 1. Implement NCP and Compute** $\beta_{\text{eff}}(t)$

**Input:** Pre-trained *source* model $\mathcal{F}_s = \{\mathcal{F}_s^{(1)}, \mathcal{F}_s^{(2)}\}$ with bottom $\mathcal{F}_s^{(1)}$ and output layer $\mathcal{F}_s^{(2)}$, target dataset $D_t$, maximum epoch $T$
1: Remove $\mathcal{F}_s^{(2)}$ from $\mathcal{F}_s$ and add on top of $\mathcal{F}_s^{(1)}$ an NCP unit $\mathcal{U}$ with multiple layers (Fig. 1b)
2: Randomly initialize and freeze $\mathcal{U}$
3: Train *target* model $\mathcal{F}_t = \{\mathcal{F}_s^{(1)}, \mathcal{U}\}$ by fine-tuning $\mathcal{F}_s^{(1)}$ on $D_t$ for epochs of $T$
4: Obtain $P$ from $\mathcal{U}$ according to Eq.(7)
5: Compute $\beta_{\text{eff}}$ with $P$ according to Eq.(2)

For an MLP $G_A$, it is possible to derive an analytical form of $\beta_{\text{eff}}$. However, it becomes extremely complicated for a deep neural network with multiple convolutional layers. To realize $\beta_{\text{eff}}$ for deep neural networks in any form, we take advantage of the automatic differentiation implemented in TensorFlow[74]. Considering the number of parameters, it is still computationally prohibitive to calculate a $\beta_{\text{eff}}$ for the entire $G_A$.

Therefore, we seek to derive a surrogate from a partial of $G_A$. Specifically, we insert a *neural capacitance probe* (**NCP**) unit, i.e., putting additional layers on top of the beheaded $G_A$ (excluding the original

output layer), and estimate the predictive ability of the entire $G_A$ using $\beta_{\text{eff}}$ of the NCP unit. Therefore, we call $\beta_{\text{eff}}$ a *neural capacitance*.

## Bayesian ridge regression

Ridge regression introduces an $\ell_2$-regularization to linear regression, and solves the problem

$$\arg\min_{\boldsymbol{\theta}} (\boldsymbol{y} - X\boldsymbol{\theta})^T (\boldsymbol{y} - X\boldsymbol{\theta}) + \lambda \parallel \boldsymbol{\theta} \parallel_2^2, \tag{9}$$

where $X \in \mathcal{R}^{n \times d}$, $\boldsymbol{y} \in \mathcal{R}^n$, $\boldsymbol{\theta} \in \mathcal{R}^d$ is the associated set of coefficients, the hyper-parameter $\lambda > 0$ controls the impact of the penalty term $\parallel \boldsymbol{\theta} \parallel_2^2$. Bayesian ridge regression introduces uninformative priors over the hyper-parameters, and estimates a probabilistic model of the problem in Eq.(9). Usually, the ordinary least squares method posits the conditional distribution of $\boldsymbol{y}$ to be a Gaussian, i.e., $p(\boldsymbol{y}|X, \boldsymbol{\theta}) = \mathcal{N}(\boldsymbol{y}|X\boldsymbol{\theta}, \sigma^2 I_d)$, where $\sigma > 0$ is a hyper-parameter to be tuned, and $I_d$ is a $d \times d$ identity matrix. Moreover, if we assume a spherical Gaussian prior $\boldsymbol{\theta}$, i.e., $p(\boldsymbol{\theta}) = \mathcal{N}(\boldsymbol{\theta}|0, \tau^2 I_d)$, where $\tau > 0$ is another hyper-parameter to be estimated from the data at hand. According to Bayes' theorem, $p(\boldsymbol{\theta}|X, \boldsymbol{y}) \propto p(\boldsymbol{\theta}) p(\boldsymbol{y}|X, \boldsymbol{\theta})$, the estimates of the model are made by maximizing the posterior distribution $p(\boldsymbol{\theta}|X, \boldsymbol{y})$, i.e., $\arg\max_{\boldsymbol{\theta}} \log p(\boldsymbol{\theta}|X, \boldsymbol{y}) = \arg\max_{\boldsymbol{\theta}} \log \mathcal{N}(\boldsymbol{y}|X\boldsymbol{\theta}, \sigma^2 I_d) + \log \mathcal{N}(\boldsymbol{\theta}|0, \tau^2 I_d)$, which is a maximum-a-posteriori (MAP) estimation of the ridge regression when $\lambda = \sigma^2/\tau^2$. All $\boldsymbol{\theta}$, $\lambda$, and $\tau$ are estimated jointly during the model fitting, and $\sigma = \tau\sqrt{\lambda}$. Based on the approach proposed by Tipping[29] and MacKay[75] to update the parameters $\lambda$ and $\tau$, we estimate $I = h(\beta_{\text{eff}}; \boldsymbol{\theta})$ with scikit-learn[76]. We can summarize the application of Bayesian ridge regression to our framework as follows:

- Inputs: $\{(\beta_{\text{eff},k}, I_k)|k = 1, 2, \ldots, K\}$ is a set of observations, where $\beta_{\text{eff},k}$ is the proposed metric calculated from the training set, $I_k$ represents the validation accuracy, $K$ is the total number of observations collected from early stage of the model training.
- Output: $I - h(\beta_{\text{eff}}; \theta) = 0$, where $\theta$ is the fitting parameters in the Bayesian ridge regression.
- Prediction: $I^* = h(0, \theta)$ as per Theorem 1.

## Reporting summary

Further information on research design is available in the Nature Portfolio Reporting Summary linked to this article.

## Data availability

Data from this study are publicly available. (1) Pre-trained ImageNet models in Keras[28], (2) Benchmark datasets CIFAR10, CIFAR100, SVHN, Fashion MNIST from Keras, (3) Kaggle challenge dataset Birds: https://www.kaggle.com/gpiosenka/100-bird-specie.

## Code availability

Code is publicly available at https://codeocean.com/capsule/6480460/tree/v1.

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

## Acknowledgements
We acknowledge the support of the USA National Science Foundation under grant #2047488, #2312501, and the Rensselaer-IBM AI Research Collaboration.

## Author contributions
C.J. and Z.H. designed experiments, conducted experiments, collected and analyzed data. T.P. conducted experiments and reported performance for baseline models. P.-Y.C. and Y.S. provided valuable insights and expertise in deep learning models. J.G. supervised the project and was the lead writer of the manuscript.

## Competing interests
The authors declare no competing interests.
