## [Peer Review File · Nature Communications]

REVIEWER COMMENTS

Reviewer #1 (Remarks to the Author):

The main contribution of this work is to use network statistics, called network capacitance β_{eff} , to predict the validation loss curve and/or final performance of the neural network being trained.

Overall I think the paper has a lot of confusing parts.

First, the definition of β_{eff} and how it is used is quite confusing. Is β_{eff} a static quality that is only related to the topology of the network? If so why Theorem 1 holds (G_A converges and thus β_{eff} is zero)? If that's not the case, it is better to clearly specify what network quantities the adjacent matrix P depends on (e.g., network weight, backpropagated gradient, data labels, etc), to make things clearer.

From its definition, β_{eff} seems to be a scalar (i.e., a single number) that changes over time, but it was used to predict the validation accuracy \mathcal{A} (also changing over time) with a regularized linear regression (Page 6 at front), and leads to very strong performance (Fig. 2-4). This seems to be too good to be true. I am not convinced that the scalar β_{eff} , summarized from a complicated neural network, contains sufficient information to predict the accuracy \mathcal{A} . Also I don't know how can we apply "Bayesian ridge regression" on scalar (1-D) input? Can the authors elaborate on how it is done? The section about "Bayesian ridge regression" is too general to be useful.

Page 5 mentions the three-layer neural capacitance probe (NCP) unit, which is supposed to predict the performance of neural networks, yet it has nothing to do with β_{eff} ? I am really confused how the prediction works.

Second, the experiments are also inadequate. A lot of ablation study are missing, as pointed out in the following:

1. What happens if the training converges but overfits (and according to Theorem 1, $\beta_{\text{eff}} = 0$), but the validation accuracy is high? Does β_{eff} still gives strong performance? The authors should do studies at different training regions, otherwise the conclusion is misleading.

2. If β_{eff} contains only the training information, there always exists a validation set so that the prediction fails. Therefore, the prediction accuracy should critically depend on some sort of distances between the training and the validation distribution. Such a study is also missing.

3. β_{eff} seems to also contain the training labels info, in addition to the weights of the network (otherwise there is no way to know when the training converges). In that case, a baseline would be just to use training accuracy (as a time series) to predict validation accuracy β_{eff} .

4. What are the other baselines shown in Table 1? What are BSV, LSV? Are there any references and how did they computed? If they are computed based on the network weights only (but not the labels in the training set), then it is not a fair comparison. Please clarify. Also please provide a conceptual comparison between the proposed method and previous methods in Table 1.

Other points:

In page 3, the authors mentioned that network reduction approach is used to decouple the network system, how does it affect gradient flow in the reduced network? From Theorem 1 it is clear that β_{eff} depends on not only network weights but also backpropagated gradient.

One of the key steps in the methodology side is to linearize the dynamics (Eqn.7). How does it affect the conclusion? Is the linearized dynamics a sufficient characterization of the original nonlinear dynamics?

In Theorem 1: what do you mean by " G_A converges"? Do you mean training G_A until it hits the local optimal of the loss function? or it fits with all the labelled data? I checked supplementary materials (S4) and seems that you use the condition that gradient is zero. What happens to SGD case when the gradient is never zero?

In supplementary materials (S3): Multiple missing references.

Reviewer #2 (Remarks to the Author):

Overall, this is an interesting paper. Below, I provide some constructive feedback that should help improve the quality of the manuscript.

Key Results

This paper proposes to use network resilience metric β_{eff} as a proxy of the model accuracy. More precisely, to estimate the accuracy of a neural network, the proposed method works as follows:

- First, replace the classification head with a random-initialized neural capacitance probe.
- Then, train the model for several epochs on the target dataset with the weight in probe frozen. For each epoch, compute the resilience metric β_{eff} based on the second-order gradient of the weights, and collect the validation accuracies.
- Finally, fit a linear model with input β_{eff} and output the collected validation accuracy. The bias of the linear model is the predicted accuracy.

The authors also provide a proof that links the proposed metric and convergence of deep networks. The predicted accuracies on the selected models appear to be highly correlated with the actual accuracies.

Validity

The authors show results on vision tasks with some CNN models. However, the number of sampled neural networks is too small to support their main claims. Standard NAS benchmark should be used for evaluation.

Significance

The significance of this work is hard to judge due to the limited benchmark selection and the lack of direct comparisons with other related approaches. More precisely, the authors should provide side-by-side comparisons with SOTA. A few examples are as follows:

[1] Lin, Ming, et al. "Zen-nas: A zero-shot nas for high-performance image recognition." Proceedings of the IEEE/CVF International Conference on Computer Vision. 2021.

[2] Mellor, Joe, et al. "Neural architecture search without training." International Conference on Machine Learning. PMLR, 2021.

[3] Tanaka, Hidenori, et al. "Pruning neural networks without any data by iteratively conserving synaptic flow." Advances in neural information processing systems 33 (2020): 6377-6389.

[4] Chen, Wuyang, et al. "Deep architecture connectivity matters for its convergence: A fine-grained analysis." Advances in Neural Information Processing Systems 35 (2022): 35298-35312.

[5] Zhang, Zhihao, and Zhihao Jia. "Gradsign: Model performance inference with theoretical insights." arXiv preprint arXiv:2110.08616 (2021).

[6] Li, Guihong, et al. "ZiCo: Zero-shot NAS via inverse Coefficient of Variation on Gradients." arXiv preprint arXiv:2301.11300(2023).

[7] Patil, Shreyas Malakarjun, and Constantine Dovrolis. "PHEW: Constructing sparse networks that learn fast and generalize well without training data." International Conference on Machine Learning. PMLR, 2021.

Data and methodology

Overall, the metric proposed in this paper is novel. I suggest the following to further strengthen the paper:

- The authors should give some toy example(s) to illustrate how exactly their proposed metric gets computed step by step. Currently, Fig. 1 only delivers the high-level idea, so more details are needed.
- Although it's left for future work, I think it is necessary to use standard benchmarks for neural architecture search (NAS), e.g., NASBench201, for results reproducibility and easier comparison with other approaches.
- GPU-hour spent for estimating the accuracy is necessary for comparison with existing works. This should include the time spent on model warmup epochs (training epochs before the starting epoch t_0), data collection epochs, and metric computation.
- Comparison with more related work is necessary. For example, reference [6] above provides a comprehensive survey on existing works on Zero-shot NAS, which is referred as TM in paper.

Analytical approach

The reason why a high network resilience may indicate a high validation accuracy is not clearly specified. Also, the proposed metric is only proved that it can indicate that the network achieves the optimal point. It would be better if the authors show some theoretical results w.r.t the convergence rate before reaching the optimum. Finally, the relationship between the generalization capacity and the proposed metric can also make this paper even stronger.

Suggested improvements

- Use publicly available NAS benchmarks for evaluation, particularly large datasets like ImageNet, Places365, and provide direct comparisons with existing approaches (SOTA).
- Provide results on more diverse CNN. Currently, the only difference among selected CNNs are their depth and basic blocks. The authors should consider more diverse CNNs, e.g., Wide-Resnet.
- Report GPU-hours as an estimation of the computational cost.
- Provide more details on the relationship between resilience and accuracy.

- [Minor] Can the author show some results beyond the vision, such as language tasks?

Clarity and context

Comparison with related works is necessary. Also, authors need to improve the clarity of the proposed method. In particular, the construction of the metric (Eq. 2) shows that the metric is determined by the adjacency matrix P . As shown in Eq. 1, g_{ij} is the weight for the edge, while the adjacency matrix P only contains 0 or 1. However, this is not the case in Eq. 2 as the authors claim that the graph is a weight graph.

Authors' Response to Reviews of

Network Properties Determine Neural Network Performance

Chunheng Jiang, Tejaswini Pedapati, Zhenhan Huang, Pin-Yu Chen, Yizhou Sun, Jianxi Gao
Nature Communication, NCOMMS-23-35278

RC: Reviewers' Comment, AR: Authors' Response, □ Manuscript Text

AR: We express our gratitude to the editor and the esteemed reviewers of Nature Communications for their valuable insights and thoughtful comments. Below, we provide detailed responses to the reviewers' comments, along with a summary of the revisions we implemented in accordance with their suggestions. We are confident that these revisions align with the high publication standards set by Nature Communications.

Reviewer #1

RC: *The main contribution of this work is to use network statistics, called network capacitance β_{eff} , to predict the validation loss curve and/or final performance of the neural network being trained. Overall I think the paper has a lot of confusing parts.*

AR: We apologize if certain sections of our previous manuscript were unclear. Taking into account your valuable comments, we have made substantial improvements to enhance clarity in the revised manuscript. We thank the Reviewer for these great questions that significantly improve the readability and clarify our manuscript. We sincerely hope that the refined manuscript meets your expectations, and we seek your agreement for its publication.

RC: *First, the definition of β_{eff} and how it is used is quite confusing. Is β_{eff} a static quality that is only related to the topology of the network? If so why Theorem 1 holds (G_A converges and thus β_{eff} is zero)? If that's not the case, it is better to clearly specify what network quantities the adjacent matrix P depends on (e.g., network weight, backpropagated gradient, data labels, etc), to make things clearer.*

AR: We would like to clarify that β_{eff} is not a static quality, but rather a scalar metric derived from both the topology and weights of the network. Although the topology is static, weights change during the training process, leading to the changes in β_{eff} . More specifically, we aimed to study the training dynamics over the weights of the neural network G_A . To facilitate our study with the well-developed techniques in network science, we modeled the training dynamics with a directed network G_B , where each node is related to a weight in G_A , and each edge is related to the interaction between a pair of weights in G_A . If the nodes' states in G_B reach equilibrium, it indicates that the weights of G_A become stable and may reach the optimal. The adjacency matrix P is a "weighted" matrix over G_B , where each entry in P is associated with an edge of G_B and describes the interaction strength between a pair of weights in G_A . The reviewer is correct that this matrix P depends on network weight, gradients, data labels, etc. We have added more descriptions in the revised manuscript.

We add the following description to the revised manuscript (first line in Section "Property of the neural capacitance"):

"According to Eq.(7), we have the weighted adjacency matrix P of G_B in place. The matrix P encodes rich information of the network, such as the topology, the weights, the gradients, and the training labels indirectly."

RC: *From its definition, β_{eff} seems to be a scalar (i.e., a single number) that changes over time, but it was used to predict the validation accuracy I (also changing over time) with a regularized linear regression (Page 6 at front), and leads to very strong performance (Fig. 2-4). This seems to be too good to be true. I am not convinced that the scalar β_{eff} , summarized from a complicated neural network, contains sufficient information to predict the accuracy I . Also I don't know how can*

we apply "Bayesian ridge regression" on scalar (1-D) input? Can the authors elaborate on how it is done? The section about "Bayesian ridge regression" is too general to be useful.

AR: The reviewer’s interpretation of β_{eff} is correct. β_{eff} is a 1D metric that measures the performance of a neural network. It is a highly compressed metric that includes the topology, weights, and gradients of the neural network. As the training progresses, both β_{eff} and the validation accuracy I change over time. β_{eff} approaches zero as the training converges. This is one novel discovery of this work.

It is not our intention to use β_{eff} to completely reflect all the impacts during training. For example, the learning rate, the optimizer, and many other hyper-parameters can affect β_{eff} values. We are leveling up the validation accuracy collected from the early training epochs to capture these missing impacts. We formulate the implicit relation between the validation accuracy and the proposed β_{eff} with the Bayesian ridge regression from the early training epochs. The relation is essentially a nonlinear function $I - h(\beta_{\text{eff}}; \theta) = 0$. It describes the training trajectory from a new perspective and is expected to be consistent throughout the training process. As the model converges, β_{eff} approaches zero (Theorem 1), we have $I^* - h(0; \theta) = 0$. Now we can directly derive from the nonlinear function a validation accuracy $I^* = h(0; \theta)$, which is exactly the final accuracy. The main purpose is to obtain the final accuracy with fewer training epochs.

Our approach aligns with existing learning curve prediction approaches, which seek to learn a non-linear predictor of validation accuracy. The prediction accuracy increases if they observe for a long enough period of time with more data points from the training trajectory. In practice, our β_{eff} can also be used together with other metrics for performance prediction. However, our approach can achieve accurate prediction even with very few early training epochs. As the reviewer stated, “This seems to be too good to be true.” We also had the same feeling when we saw these results. We confirmed these surprisingly good results when we proved Theorem 1 and conducted extensive simulations to validate it.

Regarding the use of Bayesian ridge regression, we described in Fig. 2 (row 3) in the main manuscript how to use it to estimate the relation between β_{eff} and I with a few observations on the training trajectory. We would like to emphasize that Bayesian ridge regression is applied to the observed pairs of training β_{eff} and validation accuracy rather than 1D scalar input. For clarification, we added some extra notations to section **Bayesian ridge regression in Methods**:

We can summarize the application of Bayesian ridge regression to our framework as follows:

- Inputs: $\{(\beta_{\text{eff},k}, I_k) | k = 1, 2, \dots, K\}$ is a set of observations, where $\beta_{\text{eff},k}$ is the proposed metric calculated from the training set, I_k represents the validation accuracy, K is the total number of observations collected from early stage of the model training.
- Output: $I - h(\beta_{\text{eff}}; \theta) = 0$, where θ is the fitting parameters in the Bayesian ridge regression.
- Prediction: $I^* = h(0, \theta)$ as per Theorem 1.

As an addition to the above notations and **Fig 1 (c)** in the main manuscript, we included an extra diagram on the right side to illustrate how we apply the Bayesian ridge regression in predicting the final accuracy. The orange dots are observations collected from the early training stage and used to fit the relation between the training $\beta_{\text{eff},k}$ s and validation accuracy I_k s, and then to extrapolate the final accuracy when $\beta_{\text{eff}} \rightarrow 0$, which we believe will be held when the model is converged. To be noted that some of the very early epochs may be noisy and are discarded from the regression. The number of observations and the start epoch of the observation window are determined by the Bayesian information criterion (BIC). See the last sentence of subsection "Neural network model selection with the neural capacitance β_{eff} " in the main manuscript for details.

Figure 1: Application of Bayesian ridge regression to accuracy prediction

RC: *Page 5 mentions the three-layer neural capacitance probe (NCP) unit, which is supposed to predict the performance of*

neural networks, yet it has nothing to do with β_{eff} ? I am really confused how the prediction works.

AR: The NCP unit is placed on top of the neural network G_A to replace the original output layer. One of the main purposes of the NCP unit is to calculate a surrogate β_{eff} because calculating β_{eff} for the entire network with all weights in G_A is computationally prohibitive. We are transferring the knowledge learned from ImageNet to new data sets. The bottom layers of the neural network represent low-level features of the images, and the NCP unit captures the high-level, determinant features for classification. Because of this role, we calculate the partial β_{eff} , and it is still able to predict the performance of the entire network.

We add revision in the revised manuscript (last paragraph of Section "Property of the neural capacitance"):

"Because of this, we seek to derive a surrogate from a partial of G_A . Specifically, we insert a neural capacitance probe (NCP) unit . . ."

RC: *Second, the experiments are also inadequate. A lot of ablation study are missing, as pointed out in the following:*

AR: We appreciate the reviewer’s insightful reviews and valuable suggestions regarding ablation studies. In response, we have incorporated supplementary experiments and expanded our descriptions in the revised manuscript to enhance clarity. We are confident that these changes contribute significantly to improving the paper.

RC: *1. What happens if the training converges but overfits (and according to Theorem 1, $\beta_{\text{eff}} = 0$), but the validation accuracy is high? Does β_{eff} still gives strong performance? The authors should do studies at different training regions, otherwise the conclusion is misleading.*

AR: There may be a misunderstanding. The proposed approach does not intend to predict the capacity of the neural network without considering the impacts of various factors (e.g., hyper-parameters) on the final performance of the model. Overfitting happens when the model learns the training data too well (and may only memorize the seen examples without essentially learning the knowledge from them), yet it is unable to generalize to new data. This phenomenon becomes apparent when the training accuracy continues to rise while the testing accuracy starts to decline. In the presence of overfitting, the testing/validation accuracy typically is lower than the training accuracy.

The degree of overfitting and the overall performance of a model is influenced by a variety of factors, and properties related to β_{eff} may also be distorted.

Figure 2: Training top 1 error (circles) versus validation top 1 error (squares) with a dropout rate of $p = 0.4$ (red) or $p = 0.1$ (blue) for ResNet18 (left) and ResNet34.

In response to the reviewer’s suggestion, we aimed to make an ablation study comparing the predicted final accuracy with and

Figure 3: Training top 1 error (circle) versus validation top 1 error (squares) for different batch size: 16, 32, and 64. Left figure is ResNet18 model and right figure is ResNet34 model.

without addressing overfitting. Given that overfitting is primarily induced by model complexity, and in our case, both the retrained model and NCP remain fixed, the model’s complexity remains unchanged. However, since the dropout rate in NCP is typically employed to mitigate overfitting, we attempted to reproduce an overfitting scenario by reducing the dropout rate from 0.4 to 0.1. Regrettably, we did not observe overfitting under these conditions, as shown in Figs. 2, none of the models exhibit overfitting. One of the contributing factors is that the pre-trained models may have been well-trained and reside in a favorable position within the optimal loss basin.

In addition to apply different dropout rate, we also examined the effect of different batch sizes. The result is shown in Figs. 3. We did not observe overfitting neither using different batch size.

RC: *2. If β_{eff} contains only the training information, there always exists a validation set so that the prediction fails. Therefore, the prediction accuracy should critically depend on some sort of distances between the training and the validation distribution. Such a study is also missing.*

AR: We are grateful for the reviewer’s insightful feedback. The metric β_{eff} is built on the adjacency matrix P , which encodes the topology information, the training information, the weights of the network, as well as the gradients information. Since the weights are updated during training to reduce the discrepancy between the labels and the predictions, we admit that β_{eff} also indirectly contains the training labels information. We agree with the reviewer that there always exists a validation set to fail the prediction and the prediction accuracy of our approach depends on the distribution distance between the training and validation distribution. We would like to emphasize that the underlying assumption of the proposed approach aligns with the implicit assumption of all supervised learning approaches. The statements are built on a fundamental assumption of all supervised learning approaches, i.e., the training and validation data are drawn from the same distribution. If the data is skewed, it will be difficult to rely on the validation accuracy to select the best-trained model or to make reliable predictions. We used β_{eff} from the training set to predict the validation accuracy, which essentially bridges the distribution shift. More epochs of observations are shown to improve the prediction quality.

RC: *3. β_{eff} seems to also contain the training labels info, in addition to the weights of the network (otherwise there is no way to know when the training converges). In that case, a baseline would be just to use training accuracy (as a time series) to predict validation accuracy I.*

AR: We thank the reviewer for suggesting this baseline, which is exactly one of our baseline training curve predictors. We have two learning curve predictor baselines, BSV and LSV, to predict the final validation accuracy. Instead of using the training accuracy, both baselines use the validation accuracy values collected during training to predict the final validation accuracy.

BSV uses the best validation accuracy seen so far, and LSV uses the last validation accuracy seen as the prediction of the final accuracy.

RC: *4. What are the other baselines shown in Table 1? What are BSV, LSV? Are there any references and how did they computed? If they are computed based on the network weights only (but not the labels in the training set), then it is not a fair comparison. Please clarify.*

AR: Table 1 compares our proposed method to four learning curve predictors (including two heuristic rules BSV and LSV, and two advanced methods BGRN and CL), and three transferability metrics NCE, LEEP, and LogME. The conceptual differences between these approaches are discussed in Section "Comparison with other approaches" from P8 to P9:

“... our approach has access to some observations collected from early training, and therefore our prediction mechanism is more similar to learning curve prediction than those TM-based approaches which are designed as a surrogate of the transferability without fine-tuning or re-training.”

In terms of fairness, the proposed metric β_{eff} , BSV, and LSV do not directly access the training labels. BSV uses the best-observed validation accuracy value, while LSV uses the last observed validation accuracy value as a prediction of the final accuracy [2, 8].

RC: *Also please provide a conceptual comparison between the proposed method and previous methods in Table 1.*

AR: We have mentioned the conceptual comparison between the proposed method and previous methods in Section "Comparison with other approaches" (Page 9). The key points can be summarized as follows:

Two families of previous predictors are considered in our comparison analysis: transferability measures (TMs) and learning-curve-based predictors (LCPs). TMs are proposed to quantitatively estimate how easy it is to transfer knowledge learned from a source task to a target task. The idea of LCPs is to extrapolate the partial learning curve using a combination of continuously increasing basic functions. Our approach also accesses partial observations of the learning curve collected from early training, and therefore it is more similar to LCPs than TMs.

RC: *Other points: In page 3, the authors mentioned that network reduction approach is used to decouple the network system, how does it affect gradient flow in the reduced network? From Theorem 1 it is clear that β_{eff} depends on not only network weights but also backpropagated gradient.*

AR: The universal network reduction approach (GBB reduction)[1] has been widely used in many real-world networks[7, 6, 4, 5]. One of the core techniques is mean-field theory. In short, the interactions between a node and all other nodes can be approximated by the interaction between the node and a "virtual" super-node that represents all other nodes. The N-dimensional states of the nodes in the system can be captured by a 1D effective state x_{eff} as $f(x_{\text{eff}}) + \beta_{\text{eff}}g(x_{\text{eff}}, x_{\text{eff}}) = 0$, and the equilibrium states of the system can therefore be solved from $f(x_i^*) + (P * \mathbf{1})g(x_i^*, x_{\text{eff}}) = 0$. The metric β_{eff} measures the resilience of the network, and it depends on the network weights as well as the gradients.

The reduction approach derives a powerful 1D metric to measure the global property of the network, but it does not change the gradient flow in the reduced network at all.

RC: *One of the key steps in the methodology side is to linearize the dynamics (Eqn.7). How does it affect the conclusion? Is the linearized dynamics a sufficient characterization of the original nonlinear dynamics?*

AR: Linearizing the dynamics allows us to analyze the behavior of neural networks in a more straightforward way. The motivation for linearizing the dynamics is to reformulate the training dynamics in the same form as the general dynamics described in Eqn. 1, and to analyze the behavior of neural network training. The general dynamics is characterized by three components: a self-driving force $f(\cdot)$, an external driving force $g(\cdot, \cdot)$ and an adjacency matrix P (Eqn. 1). To obtain the adjacency matrix P , we decouple the nonlinear gradient by applying the linearization at W^* .

The linearized dynamics is a sufficient characterization of the original nonlinear dynamics: $dW^{(\ell)}/dt$ is a function of $W^{(\ell+1)}$ instead of approximation of it, and $W^{(\ell+1)}$ is an explicit term in Eqn. 8.

The derivation comes from the property of backpropagation, and there is no approximation. As described in Eqs. 8 and 9, $W^{(\ell+1)}$ contributes a direct impact on the gradient, and the other weights in higher layers also affect it, implicitly and indirectly via $W^{(\ell+1)}$. When doing backpropagation, the weights on the higher layers are frozen and their impacts are propagated backwardly to $W^{(\ell+1)}$ when updating the weights $W^{(\ell)}$. These indirect impacts on the gradient are fully considered in our analysis.

RC: *In Theorem 1: what do you mean by " G_A converges"? Do you mean training G_A until it hits the local optimal of the loss function? or it fits with all the labelled data? I checked supplementary materials (S4) and seems that you use the condition that gradient is zero. What happens to SGD case when the gradient is never zero?*

AR: The convergence of G_A is established on zero gradients, a practice commonly employed in analyzing complex systems. The underlying dynamical system, as defined in Eqn. 1, attains convergence when the entire system reaches an equilibrium state. This equilibrium state is directly associated with the weight values of the neural network, where the gradients of these weights approach zero. Even though the condition of zero gradients may never occur, the theoretical analysis simplification is still effective.

RC: *In supplementary materials (S3): Multiple missing references.*

AR: We apologize for any missing references in the manuscript. This is due to the use of cross-references between the main and supplementary materials.

Correction in the revised manuscript:

- S3, 1st line: The right hand side (RHS) of Eq.(5) is a function of . . .
- S3, 3rd to the last line: The system can be viewed as a realization of the general Eq.(1), with linear . . .

Reviewer #2

RC: *Overall, this is an interesting paper. Below, I provide some constructive feedback that should help improve the quality of the manuscript.*

Key Results: this paper proposes to use network resilience metric β_{eff} as a proxy of the model accuracy. More precisely, to estimate the accuracy of a neural network, the proposed method works as follows:

- *First, replace the classification head with a random-initialized neural capacitance probe*
- *Then, train the model for several epochs on the target dataset with the weight in the probe frozen. For each epoch, compute the resilience metric β_{eff} based on the second-order gradient of the weights and collect the validation accuracies.*
- *Finally, fit a linear model with input β_{eff} and output the collected validation accuracy. The bias of the linear model is the predicted accuracy.*

The authors also provide a proof that links the proposed metric and convergence of deep networks. The predicted accuracies on the selected models appear to be highly correlated with the actual accuracies.

AR: We appreciate the reviewer’s thoughtful summary of our key results and are pleased that it is perceived as interesting. We have carefully considered your constructive comments and made the required revisions accordingly. We believe that the updated manuscript aligns with the standards for publication in Nature Communications.

RC: **Validity: The authors show results on vision tasks with some CNN models. However, the number of sampled neural networks is too small to support their main claims. Standard NAS benchmark should be used for evaluation.**

AR: We appreciate the reviewer’s comments regarding the paper’s validity. We agree with the suggestion to evaluate our approach on standard NAS benchmarks. In response, we have incorporated additional experiments using NAS-Bench-201, showcasing the improved performance of our approach. We measure and compare the ranking quality in terms of Spearman’s ranking correlation ρ of our approach with the baseline models BSV and LSV. It shows that ours can achieve $\rho = 0.76$, better than BSV’s $\rho = 0.68$ and LSV’s $\rho = 0.68$.

RC: **Significance: The significance of this work is hard to judge due to the limited benchmark selection and the lack of direct comparisons with other related approaches. More precisely, the authors should provide side-by-side comparisons with SOTA. A few examples are as follows:**

[1] Lin, Ming, et al. "Zen-nas: A zero-shot nas for high-performance image recognition." Proceedings of the IEEE/CVF International Conference on Computer Vision. 2021.

[2] Mellor, Joe, et al. "Neural architecture search without training." International Conference on Machine Learning. PMLR, 2021.

[3] Tanaka, Hidenori, et al. "Pruning neural networks without any data by iteratively conserving synaptic flow." Advances in neural information processing systems 33 (2020): 6377-6389.

[4] Chen, Wuyang, et al. "Deep architecture connectivity matters for its convergence: A fine-grained analysis." Advances in Neural Information Processing Systems 35 (2022): 35298-35312.

[5] Zhang, Zhihao, and Zhihao Jia. "Gradsign: Model performance inference with theoretical insights." arXiv preprint arXiv:2110.08616 (2021).

[6] Li, Guihong, et al. "ZiCo: Zero-shot NAS via inverse Coefficient of Variation on Gradients." arXiv preprint arXiv:2301.11300(2023).

[7] Patil, Shreyas Malakarjun, and Constantine Dovrolis. "PHEW: Constructing sparse networks that learn fast and generalize well without training data." International Conference on Machine Learning. PMLR, 2021.

AR: We thank the reviewer for bringing our attention to these related papers. We cited the suggested papers in our revised manuscript (see references 46-51) and added in Section "Comparison with other approaches":

... and many of them are training-free metrics for assessing the performance of neural networks.

Given the similarities between our approach and learning curve-based methods, it would be more fair to compare our proposed metric approach against the learning curve-based prediction approaches than these training-free metrics. However, in response to the reviewer’s highlighted reference [3], we compare our metric with the proposed one. The results of this comparison demonstrate the superiority of our approach. The training-free method ZiCo yields a Spearman’s ranking correlation of $\rho = 0.59$, significantly lower than our $\rho = 0.76$.

RC: **Data and methodology: Overall, the metric proposed in this paper is novel. I suggest the following to further strengthen the paper: The authors should give some toy example(s) to illustrate how exactly their proposed metric gets computed step by step. Currently, Fig. 1 only delivers the high-level idea, so more details are needed.**

AR: We are pleased that the reviewer recognizes the novelty of our approach. Also, we thank the reviewer for the suggestion to strengthen our paper. While Fig. 1 in the manuscript shows the high-level idea of our approach, we also have shown in

Algorithm 1: Implement NCP and Compute β_{eff} the specific steps (e.g. Step 4 and 5) on how to compute the proposed metric.

See the 1st paragraph that immediately follows Algorithm 1:

For an MLP G_A , it is possible to derive an analytical form of β_{eff} . However, it becomes extremely complicated for a deep neural network with multiple convolutional layers. To realize β_{eff} for deep neural networks in any form, we take advantage of the automatic differentiation implemented in TensorFlow.

Suppose we are now identify the interaction strength P_{ij} between weight W_i and W_j as shown in Fig. 1 (a), we need to apply the advanced automatic differentiation in Tensorflow to derive the second-order gradient

$$P_{ij} = \partial^2 \mathcal{C} / \partial W_i \partial W_j$$

over the training set (Step 4), we then apply Eq. 2 to compute the proposed metric:

$$\beta_{\text{eff}} = \frac{(\sum_j P_{1j}) \times (\sum_i P_{i1}) + (\sum_j P_{2j}) \times (\sum_i P_{i2}) + \dots + (\sum_j P_{nj}) \times (\sum_i P_{in})}{\sum_i \sum_j P_{ij}}.$$

We are confident that presenting this breakdown will aid the reviewer in gaining a thorough understanding of the computation process underlying the proposed metric.

RC: *Although it's left for future work, I think it is necessary to use standard benchmarks for neural architecture search (NAS), e.g., NASBench201, for results reproducibility and easier comparison with other approaches.*

AR: We appreciate the reviewer's suggestion to evaluate our approach on standard benchmarks for Neural Architecture Search (NAS). In response, we randomly sampled 100 neural networks from NAS-Bench-201 and assessed the ranking quality using our proposed metrics, along with two learning curve prediction approaches, LSV and BSV. The figures below demonstrate that our approach exhibits superior ranking quality compared to the baseline methods.

Figure 4: Our approach

Figure 5: BSV method

Figure 6: LSV method

RC: *GPU-hour spent for estimating the accuracy is necessary for comparison with existing works. This should include the time spent on model warmup epochs (training epochs before the starting epoch t_0), data collection epochs, and metric computation.*

AR: We appreciate the reviewer's suggestion and have incorporated the GPU-hour spent for accuracy prediction, as follows:

RC: *Comparison with more related work is necessary. For example, reference [3] above provides a comprehensive survey on existing works on Zero-shot NAS, which is referred as TM in paper.*

Table 1: Running time in GPU hours for our framework in estimating the accuracy. It involves two steps: 1) transfer learning to freeze NCP unit and then fine-tune the pre-trained model until it’s converged; 2) compute the per epoch β_{eff} s.

Model	Transfer learning	Computing β_{eff}
ResNet18	1.32	1.18
ResNet34	1.82	1.52
ResNet50	2.88	2.20
ResNet101	3.60	3.30
ResNet152	3.96	4.23

AR: We thank the reviewer for providing the reference [3], and we applied it to NAS-Bench-201 and compared it with our approach. It’s shown that our approach outperformed the proposed TM metric in [3]. As shown in Figure 3 and Figure 6, our approach’s ranking quality is $\rho = 0.76$ much better than $\rho = 0.59$ of the suggested metric in reference [3].

RC: *Analytical approach: the reason why a high network resilience may indicate a high validation accuracy is not clearly specified. Also, the proposed metric is only proved that it can indicate that the network achieves the optimal point. It would be better if the authors show some theoretical results w.r.t the convergence rate before reaching the optimum. Finally, the relationship between the generalization capacity and the proposed metric can also make this paper even stronger.*

AR: We appreciate the reviewer’s suggestion to include more theoretical findings. However, the network conversion from G_A and G_B introduced some non-smoothing operations, and the definition of the weighted adjacency matrix is built on the gradients. These conversions introduce complexity into the derivation of the coverage rate before reaching the optimum. Currently, we are still diligently exploring a feasible approach to establish the relationship between the generalization capacity and the proposed metric.

RC: *Suggested improvements*

- Use publicly available NAS benchmarks for evaluation, particularly large datasets like ImageNet, Places365, and provide direct comparisons with existing approaches (SOTA).
- Provide results on more diverse CNN. Currently, the only difference among selected CNNs are their depth and basic blocks. The authors should consider more diverse CNNs, e.g., Wide-Resnet.
- Report GPU-hours as an estimation of the computational cost.
- Provide more details on the relationship between resilience and accuracy.
- (Minor) Can the author show some results beyond the vision, such as language tasks?

AR: We appreciate the detailed suggestions by the reviewer. We have added required materials in the revised manuscript to address the comments. Here we simplify the responses with some key pointers to our revision.

Figure 7: Suggested ZiCo in reference [3]

Regarding comparison with publicly available NAS benchmarks. We added some additional experiments on NAS-Bench-201 to compare our approach with the baselines LSV and BSV as well as one additional training-free metric ZiCo. It’s found that our approach is the best on the benchmark dataset.

Regarding comparison with more diverse CNN models. We are grateful to the reviewer for bringing this to our attention. The pre-trained models reported in the manuscript encompass a broad spectrum, incorporating some of the most widely recognized neural networks. Upon promptly examining the recommended Wide-ResNet, we observed that the sole distinction between Wide-ResNet-50-2 and ResNet-50 lies in the last block— one featuring 2048-1024-2048 channels, while the other boasts 2048-512-2048 channels. Given the existing array of ResNet variations in terms of depth and width, we opted not to replicate the results for Wide-ResNet.

Regarding GPU hours. We thank the reviewer for requesting the GPU hours, so we repeated the experiment, and reported the GPU hours in Table 1.

Regarding the relationship between resilience and accuracy. We appreciate the reviewer’s suggestion to expand our theoretical findings regarding the connection between resilience and accuracy. The current theorem (Theorem 1) provides foundational insights into these relationships. However, due to the inherently non-smooth nature of the operations involved in the conversion from a neural network G_A to a line graph G_B , we acknowledge the need for further refinement. We are actively exploring more effective approaches to enhance our understanding of the intricate relationship between resilience and accuracy, aiming to surpass the limitations of our current theoretical framework.

Regarding results on language task. Our approach mainly concentrates on the vision tasks, and the complexity involved in adapting our implementation to address language tasks warrants careful consideration. As a result, we intend to explore this aspect as part of our future work.

RC: *Clarity and context: Comparison with related works is necessary. Also, authors need to improve the clarity of the proposed method. In particular, the construction of the metric (Eq. 2) shows that the metric is determined by the adjacency matrix P . As shown in Eq. 1, g_{ij} is the weight for the edge, while the adjacency matrix P only contains 0 or 1. However, this is not the case in Eq. 2 as the authors claim that the graph is a weight graph.*

AR: We appreciate the reviewer’s suggestion to improve the clarity of our approach. It is possible that the reviewer misunderstood the meaning of g_{ij} in Eqn. 1, which does not represent the weight of the edge, but describes how node j interacts with node i . The adjacency matrix P can be either a binary or a weighted matrix. In our case, the adjacency matrix is a weighted one, and the corresponding g_{ij} for the training dynamics is $w_j - w_j^*$.

References

- [1] Jianxi Gao, Baruch Barzel, and Albert-László Barabási. Universal resilience patterns in complex networks. *Nature*, 530(7590):307–312, 2016.
- [2] Aaron Klein, Stefan Falkner, Jost Tobias Springenberg, and Frank Hutter. Learning curve prediction with bayesian neural networks. In *International Conference on Learning Representations*, 2016.
- [3] Guihong Li, Yuedong Yang, Kartikeya Bhardwaj, and Radu Marculescu. Zico: Zero-shot NAS via inverse coefficient of variation on gradients. *The Eleventh International Conference on Learning Representations*, 2023.
- [4] Melanie Mitchell. Complex systems: Network thinking. *Artificial intelligence*, 170(18):1194–1212, 2006.
- [5] Márton Pósfai and Albert-Laszlo Barabasi. *Network Science*. Citeseer, 2016.
- [6] Stefan Thurner, Peter Klimek, and Rudolf Hanel. A network-based explanation of why most covid-19 infection curves are linear. *Proceedings of the National Academy of Sciences*, 117(37):22684–22689, 2020.

- [7] Nickolas M Waser and Jeff Ollerton. *Plant-pollinator interactions: from specialization to generalization*. University of Chicago Press, 2006.
- [8] Martin Wistuba and Tejaswini Pedapati. Learning to rank learning curves. In *International Conference on Machine Learning*, pages 10303–10312. PMLR, 2020.

REVIEWER COMMENTS

Reviewer #1 (Remarks to the Author):

Thanks for the authors' feedbacks. I really appreciate them.

The explanation of Bayesian Ridge regression helps a lot. Now I understand how the regression is done (by collecting samples from the early stage of training).

I have follow-up questions.

1. The authors say that the three-layer neural capacitance probe (NCP) unit network is to compute a surrogate for β_{eff} , which is computationally prohibitive. How did you train this NCP network? Did you use ground truth β_{eff} as a supervised signal? There are two possibilities:

a) If you use the ground truth signal of β_{eff} as the supervision, then is it still computational costly to train the NCP? How long will it take to do the training? Please report.

b) If you choose to train NCP directly with the final accuracy (e.g., using the output of NCP for ridge regression and do end-to-end backpropagation), then it is not clear whether the output of NCP is actually connected to β_{eff} . Please clarify.

2. When I asked about overfitting, I do not argue that the model will overfit on the real-world dataset (actually overfitting does not happen, as shown in the additional experiments). Instead, I want to see a scenario that when the model overfits and have very low validation accuracy, your approach can still predict the validation accuracy very well. Otherwise the proposed method is not tested beyond the common training region and I am not convinced. To make the model overfit, we could just use a small subset of training set and see whether the proposed approach gives good prediction of the final accuracy.

3. Thanks the authors to provide the backgrounds of the baseline methods that also predict the final performance given the early stage training. There may exist other straightforward baselines, e.g., predicting the validation accuracy from the name of the dataset, from training hyperparameters (e.g., lr, length of epochs, batchsize, #parameters, etc) that are not depending on the actual training dynamics (e.g., changing weights of the networks). We would like to see whether β_{eff} actually make a difference here, which is the main contribution of the paper.

Reviewer #1 (Remarks on code availability):

See the comment above.

Reviewer #2 (Remarks to the Author):

This reviewer appreciates authors' effort to revise the initial manuscript and address individual comments. While this revised version addresses some of the initial concerns, a few important ones need further attention:

1. Comparison with SOTA

- First, the comparison against ZiCo needs to provide all the details of the setup used; without these details, it's impossible to understand the significance of the comparison and enable reproducibility of the results.
- Second, comparison against ZiCo only is not enough; comparisons against another approaches are needed to see the consistency of the results. At the very minimum, comparisons against Zen-Nas and NASWOT (refs [1] and [2] in the Reviewer#2 initial comment) are needed, with all details given in the Supplementary, to understand how this approach fares compared to existing approaches.
- Third, authors comment about the superiority of their approach compared to ZiCo needs to be rephrased since their approach involves training, hence it is expected to be provide better accuracy compared to zero-shot approaches. On the other side, zero-shot approaches have their inherent advantages so it's hard to talk about superiority in such a context.

2. Including authors relevant responses in the main manuscript/Supplementary

Some of author responses to this reviewer questions need to be included in the main manuscript and/or Supplementary, as appropriate; this is critical for improving the paper readability and overall contribution. As of now, very little material from authors responses is actually included in the revised version of the initial paper. For instance, the comparisons against SOTA required above, as well as plots of Figs.4-6 discussed in authors response need to be included in the Supplementary. Same about Table 1 (GPU times). On the other hand, authors comment regarding the relationship between resilience and accuracy should be mentioned in the main paper as current limitation.

Authors' Response to Reviews of

Network Properties Determine Neural Network Performance

Chunheng Jiang, Tejaswini Pedapati, Zhenhan Huang, Pin-Yu Chen, Yizhou Sun, Jianxi Gao
Nature Communication,

RC: Reviewers' Comment, AR: Authors' Response, □ Manuscript Text

First of all, we would like to express our sincere gratitude for both reviewers' invaluable contributions that further improve our manuscript. Their dedication, expertise and conscientiousness have significantly enriched our work. We are grateful for the opportunity to benefit from both reviewers' insights.

1. Reviewer #1

RC: *Thanks for the authors' feedbacks. I really appreciate them. The explanation of Bayesian Ridge regression helps a lot. Now I understand how the regression is done (by collecting samples from the early stage of training). I have follow-up questions.*

AR: We are glad that our explanation was helpful and we believe that the revision addressed all the follow-up questions of Reviewer #1.

RC: *a) If you use the ground truth signal of β_{eff} as the supervision, then is it still computational costly to train the NCP? How long will it take to do the training? Please report.*

AR: **Whether we use the ground truth signal of β_{eff} as the supervision:** In our supervised learning scheme, we do not use β_{eff} as the ground truth. The supervised learning is used to train a new model consisting of the pretrained model and NCP layers. β_{eff} is computed by the training dynamics of the supervised learning process. In the prediction of model performance (*i.e.* accuracy), we use Bayesian regression of accuracy and β_{eff} to predict the performance at converge ($\beta_{\text{eff}} = 0$). So β_{eff} is not used in the supervised fashion when training a model.

Computational cost of our method: The direct way to get the model performance is to train the model to converge. Let the total number of epochs be K . $T_{\beta_{\text{eff}}}$ denotes the time to compute the β_{eff} and T_{train} denotes the time to train the model for each epoch. In our method, we only need to train the model for k epochs, where $k \ll K$. Therefore, our total time cost is $T_{\text{ours}} = k \times (T'_{\text{train}} + T_{\beta_{\text{eff}}})$ while the direct way (full training) is $T_{\text{full}} = K \times T_{\text{train}}$. Note that T'_{train} is close to T_{train} as we only use 3-layer NCP and its weight is frozen. Since $T'_{\text{train}} + T_{\beta_{\text{eff}}}$ is larger than T_{train} by a small margin but $k \ll K$, our computational cost is mild. In our manuscript, we show that we can predict the final accuracy by observing only $k = 10$ of $K = 100$ full training epochs, and T_{ours} is only 13% of T_{full} . The details regarding the analysis of the computation cost are stated in the section **Running time analysis** of the main page.

Our approach is efficient, especially for large and deep neural networks. Different from the training task that involves a full FP and BP . . .

The running time in GPU hours for predicting model performance on CIFAR-10 dataset is included in the updated supplementary materials (section title is **COMPUTATIONAL COST OF THE PROPOSED FRAMEWORK**).

RC: *b) If you choose to train NCP directly with the final accuracy (e.g. using the output of NCP for ridge regression and do end-to-end backpropagation), then it is not clear whether the output of NCP is actually connected to β_{eff} . Please clarify.*

AR: **Whether we train NCP directly with final accuracy:** The lightweight NCP unit plays a dual role: 1) replacing the original

Figure 1: Schematic illustration of Bayesian regression

top layers of the pretrained model for transfer learning, and 2) computing the metric β_{eff} as well as the validation accuracy to gather some observations for downstream Bayesian regression.

For 1), we randomly initialize NCP and freeze it during the fine-tuning of the neural network. Since the NCP remains frozen, the fine-tuning process does not involve any training for the NCP. For 2), NCP consists of only 3 layers, which is much smaller than the entire neural network. As highlighted in the main manuscript (see the last two paragraphs of "Neural capacitance" in Methods), computing β_{eff} for the entire network is prohibitively expensive, so we seek to compute the β_{eff} with NCP.

If the reviewer's mention of 'training' refers to the Bayesian regression model, the NCP's role is to collect the encoded signals from bottom layers of the entire neural network, enabling the computation of observations, i.e., β_{eff} and validation accuracy. These observations are then utilized by the Bayesian regression model to learn the implicit relationship between β_{eff} and validation accuracy. The ultimate goal is to apply Theorem 1 on this Bayesian regression model, extrapolating the final accuracy. Obtaining the final accuracy of the neural network without our framework typically requires training the model until it converges. Our framework saves the extra work. Despite the absence of any training on the NCP, we still can predict the neural network's final accuracy based on our framework.

We hope the above explanation clarifies the relations between NCP and β_{eff} , and β_{eff} and the final accuracy. In the main paper, we illustrate the proposed algorithm as the algorithm shown below:

Algorithm 1 Implement NCP and Compute β_{eff}

Input: Pre-trained *source* model $\mathcal{F}_s = \{\mathcal{F}_s^{(1)}, \mathcal{F}_s^{(2)}\}$ with bottom $\mathcal{F}_s^{(1)}$ and output layer $\mathcal{F}_s^{(2)}$, target dataset D_t , maximum epoch T

- 1: Remove $\mathcal{F}_s^{(2)}$ from \mathcal{F}_s and add on top of $\mathcal{F}_s^{(1)}$ an NCP unit \mathcal{U} with multiple layers (Fig. 1b)
 - 2: Randomly initialize and freeze \mathcal{U}
 - 3: Train *target* model $\mathcal{F}_t = \{\mathcal{F}_s^{(1)}, \mathcal{U}\}$ by fine-tuning $\mathcal{F}_s^{(1)}$ on D_t for epochs of T
 - 4: Obtain P from \mathcal{U} according to Eq. (7)
 - 5: Compute β_{eff} with P according to Eq. (2)
-

RC: *When I asked about overfitting, I do not argue that the model will overfit on the real-world dataset (actually overfitting does not happen, as shown in the additional experiments). Instead, I want to see a scenario that when the model overfits and have very low validation accuracy, your approach can still predict the validation accuracy very well. Otherwise the proposed method is not tested beyond the common training region and I am not convinced. To make the model overfit, we could just use a small subset of training set and see whether the proposed approach gives good prediction of the final accuracy.*

AR: This is a great suggestion! We examine the performance of the proposed framework when using a subset of the training set.

We test the performance using about 0.5% of the original training data (batch size is 64 and the number of minibatches is 4). The subset of the training data is randomly sampled and each training epoch uses the same subset of the training dataset. We use the exactly same hyperparameters to fine-tuning the model, compute β_{eff} , and predict the model performance. Accuracy is predicted on CIFAR-10 dataset for ResNet50, ResNet34, ResNet18, ResNet152, ResNet101, DenseNet201, Densenet169, DenseNet161 and DenseNet121. The average test accuracy is 62.08 ± 8.25 while the average train accuracy is 79.78 ± 11.63 . We compute the Spearman correlation ρ between the test accuracy and predicted validation accuracy. The correlation is shown in the Figure 2. Due to the overfitting, the model performance is seriously degraded. But our proposed method rank model performance reasonably well.

Figure 2: Ranking correlation between the predicted performance and the real performance for models in the overfitting regime.

RC: *Thanks the authors to provide the backgrounds of the baseline methods that also predict the final performance given the early stage training. There may exist other straightforward baselines, e.g. predicting the validation accuracy from the name of the dataset, from training hyperparameters (e.g. lr, length of epochs, batchsize, #parameters, etc) that are not depending on the actual training dynamics (e.g. changing weights of the networks). We would like to see whether β_{eff} actually make a difference here, which is the main contribution of the paper.*

AR: We agree that some auxiliary information of the network and dataset can be used as a prediction. Since in our experiments we use the same number of total epochs and batch size for all models in each dataset, these two auxiliary features won't be useful for performance prediction. On the other hand, the reviewer is correct that the number of parameters is commonly used as a competitive training-free baseline in the prediction of the model performance [2]. We compared our method with the total number of parameters as a predictor. The same architectures in NAS-Bench-201 are used and we compute the Spearman correlation ρ between the number of parameters (*i.e.* model size) and the test accuracy. We find that our method has $\rho = 0.76$ while the number of parameters as metric only has $\rho = 0.52$. It is worth mentioning that the comparison is not a fair one as our method requires training but the number of parameters does not require training.

2. Reviewer #2

RC: *This reviewer appreciates authors' effort to revise the initial manuscript and address individual comments. While this revised version addresses some of the initial concerns, a few important ones need further attention:*

AR: We appreciate Reviewer #2’s valuable comments and suggestions that have significantly improved our manuscript. Although we tried to address all your comments carefully, we are sorry that the previous version is still unsatisfactory. We thank Reviewer #2 for further suggestions and have revised our manuscript accordingly.

RC: Comparison with SOTA

RC: • *First, the comparison against ZiCo needs to provide all the details of the setup used; without these details, it’s impossible to understand the significance of the comparison and enable reproducibility of the results.*

AR: We thank Reviewer #2 for the suggestion, according to which we provide the implementation details below:

Experiment details using our proposed method: We randomly sample 108 architectures in the NAS-Bench-201 search space. The highest test accuracy of the sampled architecture is 89.16%, while the lowest test accuracy is 70.09%. In the NCP framework, a three-layer NCP is added to the pretrained model in the NAS-Bench-201 search space. We use the CIFAR10 dataset as the training dataset. Each dense layer has a dimension of 128, followed by ReLU activation and batch normalization. We use exactly the same hyperparameters: the number of epochs is 50, and the dropout rate of the dropout layer in the NCP layers is 0.4. We use an SGD optimizer with a constant learning rate of 0.001. After training the new model with inserted NCP for 50 epochs, we compute the β_{eff} dynamics for each epoch. We select $[t_0, t_0 + 3]$ as the prediction window to predict the validation accuracy at the final epoch (*i.e.* $t_{\text{final}} = 50$). Bayesian regression is used for fitting, and prediction is made by extrapolation to the final epoch $|\beta_{\text{eff}}| = 0$. Figure 3 shows the typical fitting result. Using the predicted validation accuracy, we compute the correlation between the test accuracy and the predicted accuracy. The result is shown in Figure 4. The correlation in the supplementary materials indicates that our proposed framework also achieves a good prediction on the NAS-Bench-201 search space.

Experiment details using ZiCo metric: For the ZiCo method, we use a batch size of 128 and 4 minibatches. The aforementioned randomly sampled architectures are used for computing the ZiCo score. After the ZiCo score is obtained, we calculate the ranking correlation between the score and model performance.

The description is included in the supplementary materials section **COMPARISON WITH STANDARD NAS BENCHMARKS**.

Figure 3: Bayesian regression between the validation accuracy and $|\beta_{\text{eff}}|$ for the models in the NAS-Bench-201 search space. Left: typical fitting result for architectures with high accuracy. Right: typical fitting result for architectures with medium accuracy.

Figure 4: Spearman correlation of predictors and test accuracy. Left: our method. Middle: BSV method. Right: LSV method.

RC: • *Second, comparison against ZiCo only is not enough; comparisons against another approaches are needed to see the consistency of the results. At the very minimum, comparisons against Zen-Nas and NASWOT (refs [1] and [2] in the Reviewer#2 initial comment) are needed, with all details given in the Supplementary, to understand how this approach fares compared to existing approaches.*

AR: In the calculation of ZiCo [3], NASWOT [5], and ZenNAS [4] scores, we use the precisely same sampled architectures. The hyperparameters are the same as reported in the references. For the NASWOT score, we use a batch size of 128 and 1 minibatch. For ZenNAS score, we use a batch size of 16 and 32 minibatches. Following the suggestion, we included new comparisons to ZiCo, NASWOT, and ZenNAS in the revised Supplementary Information section **COMPARISON WITH STANDARD NAS BENCHMARKS**. The correlations between the NAS metric and model performance are computed using randomly sampled architectures. Using subset of architectures in the NAS-Bench-201 search space, the Spearman correlation ρ for ZiCo, NASWOT, ZenNAS is 0.59, 0.58 and -0.04, respectively. Our method has the correlation of 0.76 in comparison. The correlation for all architectures in the NAS-Bench-201 search space is 0.81 (ZiCo), 0.77 (NASWOT), and 0.35 (ZenNAS). The correlation for NASWOT and ZenNAS is reported in [2] while the correlation for ZiCo is reported in [3].

RC: • *Third, authors comment about the superiority of their approach compared to ZiCo needs to be rephrased since their approach involves training, hence it is expected to be provide better accuracy compared to zero-shot approaches. On the other side, zero-shot approaches have their inherent advantages so it's hard to talk about superiority in such a context.*

AR: We fully agree with Reviewer #2 that it is hard to talk about superiority in such a context. We rephrase our statement regarding the comparison of our method with training-free NAS methods as also listed below. The comparison is included in the updated supplementary materials.

“Our method shows a higher correlation on NAS-Bench-201 compared to ZiCo and NASWOT. It is worth mentioning that direct comparison solely on the correlation is unfair since our method is not training-free.”

RC: *Including authors relevant responses in the main manuscript/Supplementary.*

AR: Based on both reviewers’ comments in our last version, we did supplementary experiments and put them in the SI. We thank Reviewer #2 for this great suggestion. We have incorporated these new results and additional details in the SI. Specifically, here are the main places we made significant changes:

- We include the computational cost in the Supplementary Information. The section title is **COMPUTATIONAL COST OF THE PROPOSED FRAMEWORK**.

"We record the running time in GPU hours for the proposed NCP framework on CIFAR-10 dataset (image resolution is 32×32). The computational cost is shown in the Table. The NCP framework consists of two stages: in the first stage, the new model consisting of pretrained model and NCP layers with random initialization is fine-tuned for 50 epochs. The pretrained model is unfrozen while NCP layers are frozen. . . "

- We include the comparison of our method with training-free NAS methods, LSV and BSV on NAS-Bench-201 in the Supplementary Information. The section title is **COMPARISON WITH STANDARD NAS BENCHMARKS**

"In addition to examine our method on classic deep learning models such as ResNet and DenseNet, we also examine the effectiveness of the NCP method on NAS-Bench-201 [1]. . . "

References

- [1] Xuanyi Dong and Yi Yang. Nas-bench-201: Extending the scope of reproducible neural architecture search. *arXiv preprint arXiv:2001.00326*, 2020.
- [2] Arjun Krishnakumar, Colin White, Arber Zela, Renbo Tu, Mahmoud Safari, and Frank Hutter. Nas-bench-suite-zero: Accelerating research on zero cost proxies. *Advances in Neural Information Processing Systems*, 35:28037–28051, 2022.
- [3] Guihong Li, Yuedong Yang, Kartikeya Bhardwaj, and Radu Marculescu. Zico: Zero-shot nas via inverse coefficient of variation on gradients. *arXiv preprint arXiv:2301.11300*, 2023.
- [4] Ming Lin, Pichao Wang, Zhenhong Sun, Heseng Chen, Xiuyu Sun, Qi Qian, Hao Li, and Rong Jin. Zen-nas: A zero-shot nas for high-performance image recognition. In *Proceedings of the IEEE/CVF International Conference on Computer Vision*, pages 347–356, 2021.
- [5] Joe Mellor, Jack Turner, Amos Storkey, and Elliot J Crowley. Neural architecture search without training. In *International Conference on Machine Learning*, pages 7588–7598. PMLR, 2021.

REVIEWERS' COMMENTS

Reviewer #1 (Remarks to the Author):

The authors have addressed most of my concerns. In the next revision, it would be great for the authors to write $\beta_{\text{eff}}(t)$ to reflect that it is a quantity that changes over time, and clearly states the actual training algorithms. The current version is quite confusing to read.

Reviewer #2 (Remarks to the Author):

The authors addressed most of my previous concerns; I appreciate their effort in this directly. However, I still have two issues with this revised version:

- The authors must clarify the specific advantages of their proposed method in the context of its training-based nature, while acknowledging the distinct benefits of zero-shot NAS in terms of efficiency. As of now, there is only one such comment (i.e., training-based vs zero-shot approaches) buried deeply in the Supplementary Material so the regular reader may get the wrong impression about the advantages of the proposed method.

- Regarding the NAS-Bench-201 evaluation: I'm confused as to why the authors choose only a subset of 108 architectures instead of the entire dataset? To eliminate the potential risk for cherry-picking here, the authors should report their results on the entire dataset.

Authors' Response to Reviews of

Network Properties Determine Neural Network Performance

Chunheng Jiang, Zhenhan Huang, Tejaswini Pedapati, Pin-Yu Chen, Yizhou Sun, Jianxi Gao
Nature Communication, NCOMMS-23-35278

RC: Reviewers' Comment, AR: Authors' Response, Manuscript Text

We appreciate reviewers' contributions that further improve our manuscript and comments that enhance the interpretability of our work to readers.

1. Reviewer #1

RC: *The authors have addressed most of my concerns. In the next revision, it would be great for the authors to write $\beta_{\text{eff}}(t)$ to reflect that it is a quantity that changes over time, and clearly states the actual training algorithms. The current version is quite confusing to read.*

AR: We have modified our notation about β_{eff} to emphasize that it is a function of training epochs.

"As shown in Algorithm 1, the NCP does not involve fine-tuning and is merely used to calculate the neural capacitance $\beta_{\text{eff}}(t)$, which varies as the number of epochs t changes. To keep the notation succinct, we use β_{eff} to represent $\beta_{\text{eff}}(t)$."

2. Reviewer #2

RC: *The authors addressed most of my previous concerns; I appreciate their effort in this directly. However, I still have two issues with this revised version:*

- **The authors must clarify the specific advantages of their proposed method in the context of its training-based nature, while acknowledging the distinct benefits of zero-shot NAS in terms of efficiency. As of now, there is only one such comment (i.e., training-based vs zero-shot approaches) buried deeply in the Supplementary Material so the regular reader may get the wrong impression about the advantages of the proposed method.**

AR: We add more clarifications in the main manuscript.

"In addition to LC-based predictors, we compared our method with training-free NAS methods. The result is shown in the Supplementary section **Comparison with Standard NAS Benchmarks**. Direct comparison on the prediction performance (indicated by the ranking correlation) is not desirable since training-free NAS methods do not require training while our proposed method requires training of the model to compute β_{eff} ."

- **Regarding the NAS-Bench-201 evaluation: I'm confused as to why the authors choose only a subset of 108 architectures instead of the entire dataset? To eliminate the potential risk for cherry-picking here, the authors should report their results on the entire dataset.**

AR: The entire NAS-Bench-201 benchmark has a large number of neural networks. Even the prior arts of training-free NAS [1, 2], despite of light computation nature, compute correlation based on randomly sampled 1000 architectures. We understand the reviewer's concern. Considering the limit on computational resource and time, we increase the randomly sampled architectures and test our method. Figure 1 shows the Spearman correlation ρ using 108, 540, 1080 and 2160

architectures. Note that a small set of architectures is a subset of large set. In other words, when increasing the total number of architectures, we include the original randomly selected set. We find the ranking correlation increases as we increase the number of randomly sampled architectures.

Figure 1: Spearman correlation ρ of the proposed method on NAS-Bench-201 benchmark using different number of randomly sampled architectures.

References

- [1] Mohamed S Abdelfattah, Abhinav Mehrotra, Łukasz Dudziak, and Nicholas D Lane. Zero-cost proxies for lightweight nas. *arXiv preprint arXiv:2101.08134*, 2021.
- [2] Joe Mellor, Jack Turner, Amos Storkey, and Elliot J Crowley. Neural architecture search without training. In *International Conference on Machine Learning*, pages 7588–7598. PMLR, 2021.